

# An integrated multi-fingerprint sensitivity-nested approach for regional model parameter estimation and catchment similarity assessment

Simon Höllering[1], Jürgen Ihringer[1], Luis Samaniego[2], and Erwin Zehe[1]

[1]Karlsruhe Institute of Technology (KIT), Karlsruhe, Germany
[2]Helmholtz Centre of Environmental Research, Leipzig, Germany

*Correspondence to:* Simon Höllering (simon.hoellering@kit.edu)

**Abstract.** The present study provides a novel approach to the challenge of identifying behavioural parameters in the context of parameter sensitivity and related hydrologic similarity classification. A methodical framework is presented wherein global sensitivity analysis of a spatially distributed conceptual hydrologic model within 14 different mesoscale headwater catchments is combined with a parameter estimation scheme based upon both classification by (1) physiographic and climate and (2) related
dynamic response characteristics represented by hydrologic signatures (fingerprints) creating an interface between hydrologic variables of observed and simulated origin. Changing ranks in (3) partial parameter sensitivities within the catchments indicate that hydrologic dynamics might be governed by different hydrologic processes. Model simulated and the respective observed response fingerprints are found to cluster within typical sample regions. These findings show a general model adequacy to represent mesoscale streamflow response processes that relate temporally dominant parameters and allow a reasonable con-
straint on the parameter space. The senstivity-nested approach may be useful to calibrate hydrologic models sequentially on streamflow sections as well as on constraining (observable) single or combined hydrologic fingerprints and also to transfer results to similar sites, ungauged or anthropogenically altered.

## 1   Introduction

### 1.1   Defining hydrologic similarity

Assessment of catchment similarity is a cardinal challenge in hydrologic research. Although it might have been motivated by similarity theory in hydraulics and fluid mechanics (Monin and Obukhov, 1954), according to Dyck and Peschke (1995) it is extraordinarily more complex than its hydraulic archetype. Hydrologic similarity comprises physiographic, meteorologic, ecologic and geologic factors, notwithstanding human influences. As stressed by Oudin et al. (2010) one may approach the similarity problem from different angles. The classical regionalisation approach aims to represent similar catchments by similar
behavioural parameter sets within a hydrologic model (Hirsch, 1979; Castellarin et al., 2001; Götzinger and Bárdossy, 2007; Patil and Stieglitz, 2012). The proposed measures for catchment similarity in related studies range from spatial proximity (Patil and Stieglitz, 2012; Vandewiele and Elias, 1995), which is often a good predictor for a similar climate setting, to similarity





of various physiographic catchment properties (Parajka et al., 2005) or combinations thereof (Merz and Blöschl, 2005; Oudin et al., 2008). Physiographic catchment similarity is, except with respect to geomorphic descriptors, often characterised by categorical data on soil, landuse and geology. These data sources are frequently translated into metric catchment descriptors by means of their areal share (e.g. Ali et al. (2012); Merz and Blöschl (2003); Hundecha and Bárdossy (2004); He et al. (2011)) and

related to behavioural model parameters, for instance by means of variance reduction (He et al., 2011) or multiple regressions (Merz, 2006), which can be further constrained by a Lippschitz condition (Hundecha and Bárdossy, 2004; Götzinger and Bárdossy, 2007). Establishing a link between observable physiographic catchment properties and parameters of conceptual hydrologic models (CHMs) requires an extended calibration-oriented context. Correlation analysis, including model parameters and their estimation or even transfer to ungauged sites, is generally complicated by uncertainty inherent in parameters of CHMs

(Bárdossy, 2006; Fenicia, 2008) and equifinality in acceptable model results obtained by different parameter sets (Beven and Freer, 2001; Beven, 2006). For this reason, Bárdossy (2006) proposed to use vectors containing pre-selected parameter sets for a reasonable model calibration in ungauged catchments.

However, parameter regionalisation feasibly represents "similar" catchments by similar parameters in conceptual models, this method of defining similarity/dissimilarity is restricted to those differences that can still be represented by the chosen

model structure. This might imply that two catchments are treated as similar in a semi-lumped model structure, but will appear as dissimilar in a fully distributed model structure.

Alternatively, Wagener et al. (2007) proposed to distinguish similarity with respect to climate and physiographic catchment characteristics from dynamic response measures characterising streamflow release from or water storage within catchments.

The ultimate vision is to relate similarity of independent properties, sorted in a multidimensional space, and dependent functional catchment descriptors in regionally specific classification schemes.

## 1.2 Classifying similar catchments

A generally accepted classification system in hydrology is defined as a *crucial step towards hydrologic synthesis* (Ali et al.,

2012), a *fundamental step towards improved catchment hydrology science* (He et al., 2011) or as an *important organising principle, complementing the concept of the hydrologic cycle with the principle of mass conservation* (McDonnell and Woods, 2004) or even with energetic optimality (Zehe et al., 2013; Kleidon et al., 2013). The feasibility and usefulness of any classification scheme depends crucially on a) the identification of similarity measures/fingerprints of catchment behaviour and b) a successful explanation of differences in hydrologic functioning by differences in the catchment climate and physiographic

setting. In this context McDonnell and Woods (2004) proposed dimensionless numbers, dominant hydrologic processes or functional traits as assortative elements.

A well known and successful classification system is the Hydrology of Soil types (HOST) in the United Kingdom (Boorman et al., 1995). It builds on the insight that the physical properties of soils and particularly soil structure, crucially influence





catchment hydrologic functioning in this region. Another regionally specific scheme, is the River Environment Classification (REC) in New Zealand (Snelder et al., 2004), which is based on the insight that catchment climate, topography, geology and land cover are decisive for riverine ecology. Winter (2001) proposed that Fundamental Hydrologic Landscape Units (FHLU) can be defined and distinguished based on their land surface form, hydraulic properties, geologic characteristics and climatic

setting. In accordance with this Zehe et al. (2014) proposed a hierarchy of functional units that may guide experimental characterisation and development of appropriate process models for lower mesoscale catchments.

Alternatively, there have been several recent attempts to classify catchment dynamics often based on streamflow response characteristics. Sawicz et al. (2011) used six hydrologic response characteristics derived from precipitation, temperature and

streamflow data to classify 280 catchments in the eastern USA by means of a Bayesian clustering algorithm. In line with this, Carrillo et al. (2011) relates the climate setting and physiographic characteristics of several US catchments to their hydrologic behaviour by means of dimensionless numbers derived from process-based signatures. Merz and Blöschl (2003) proposed a catchment-based framework, mainly relying on the classification of climate and meteorologic controls to classify the flood generation for the whole of Austria. Gharari et al. (2011) distinguished the three hydrologic response units - wetland, hillslope

and plateau - with distinct runoff generation process (saturation excess overland flow, storage excess sub-surface flow, deep percolation) as building blocks for hydrologic modelling. To this end they adopted a landscape classification based on the Height Above Nearest Drainage proposed by Rennó et al. (2008). Di Prinzio et al. (2011) employed Self Organising Maps (SOM) in combination with a Principal Component Analysis (PCA) and Canonical Correlation Analysis (CCA) for classifying 300 Italian catchments, using indices of streamflow regime and geomorphoclimatic characteristics. In another study, also based

on SOM, Ley et al. (2011) performed a cluster analysis to identify, investigate and delineate hydrologically similar catchments considering their response behaviour in terms of more than 8200 event runoff coefficients (ERCs) and flow duration curves of 53 gauged catchments in Germany.

### 1.3    The inconconsistency of functional and physiographic similarity - a new paradox?

Notwithstanding, the appeal and success of classifying catchments into "functionally similar" groups, it is not straight forward to explain these similarities based on catchment characteristics. A comparison of structural and functional similarity indices of 36 Scottish catchments by means of the affinity propagation clustering algorithm by Ali et al. (2012) revealed, for instance, a rather inconsistent picture between flow-derived similarity and physiographic similarity indicators. This is in line with the results of several other studies on similarity metrics which e.g. concluded that the overlap of the two groups with 60 % (Oudin

et al., 2010) and 67 % respectively (Ley et al., 2011) is relatively small but can be raised e.g. by an analysis with Self-Organising Maps (SOM).

Parts of this inconsistency may be explained by the inherent equifinality and non-uniqueness of most of our governing equations (Beven, 1990; Zehe et al., 2014). Particularly, runoff as a mass flux leaving a catchment control volume is a non-unique



product of a drying potential gradient and the control volume conductance/resistance. Systems that distinctly differ with respect to the topographic controls on the driving gradients and the pedo-/geologic controls on control volume conductance may nevertheless produce runoff in a similar fashion (Wienhöfer and Zehe, 2014; Binley and Beven, 2003). Topographic and pedologic controls on runoff generation must thus be interpreted as a group, to judge how they jointly control runoff behaviour.

The necessary metric data sets on topography and soil water characteristics, are unfortunately not at hand. In line with this (Ley et al., 2011) attributes the inconsistency of functional and physiographic catchment similarity to the equifinality of runoff processes where a similar hydrologic response behaviour can be triggered by special (unique) physiographic catchment properties. Furthermore Oudin et al. (2010) invoked unique hydrologic settings of catchments and an absence of descriptors that are capable of linking the physiographic and functional catchment similarity as reasons.

### 1.4 Functional fingerprints for constraining hydrologic models

Fingerprints of catchment functioning may, furthermore, serve as additional measures to test and improve the adequacy of hydrologic models (Oudin et al., 2010), which might ultimately imply a better representation of catchment hydrologic functioning in hydrologic models. Castiglioni et al. (2010) proposed a calibration methodology for conceptual models which should

generally be both as parsimonious as possible as well as adaptable to the degree of available data in the catchment. For their spectral, multiple objective calibration they suggested the use of catchment specific signatures derived as statistical river flow indices such as the standard deviation and the autocorrelation of the river flow data in order to result in an uncertainty envelope hydrograph formed by the multiple behavioural parameter sets. Yadav et al. (2007) and several other authors (Shamir et al., 2005a, b; Zhang et al., 2008) used dynamic response characteristics of watersheds to relate them to physically measurable

catchment characteristics in order to derive regression relationships to raise the information considered in the parameter transfer and finally to reasonably constrain the parameter space. Vrugt and Sadegh (2013) also preferred to use indices based on hydrologic signatures to evaluate hydrologic models instead of the commonly applied standard efficiency measures based on residuals.

## 2 Objective and driving research questions

In line with the recommendation of Vrugt and Sadegh (2013) this study explores the consistency between catchment structural and functional similarity indices and particularly the value of the latter for constraining distributed hydrologic models. Specifically, we address three main research questions to contribute to clarification within the interrelated fields of hydrologic similarity, catchment classification and objective catchment functional parameter estimation:

– Can we constrain the feasible parameter space of a distributed hydrologic model in a consistent manner using fingerprints characterising streamflow generation during high, average and low flow conditions?





- May we employ regional differences in "dependent" parameter sensitivity of a distributed model to learn about regional differences in catchment functioning?

- Can we explain differences in catchment behaviour and parameter sensitivity based on typically available physiographic and meteorologic catchment descriptors?

We employed the mesoscale Hydrologic Model (mHM) (Samaniego et al., 2010b) as state-of-the-art distributed hydrologic model and the Fourier Amplitude Sensivity Test (FAST) (Cukier et al., 1973; Reusser, 2010) to explore temporal dependent parameter sensitivity of simulated streamflow within different subcatchments of the Ruhr basin in Germany. In particular, we employed the FAST method to explore parameter sensitivity of the selected fingerprints, within the range of their independent variable.

# 3 Concept and methodological approach

## 3.1 General considerations

Our approach rests on three pillars, 1) indices characterising the landscape properties, 2) streamflow generation and 3) dependent partial parameter sensitivities of mHM of simulated streamflow and of the related fingerprints. We explicitly focus on the headwaters of the strongly regulated Ruhr basin, to analyse, so far as possible, unaffected hydrologic regimes, thereby 15 conserving the ability to extend and test the approach to more strongly altered downstream locations. Our analysis, which as implemented in the Ruhr catchment using mHM, maybe generalised to any model or catchments using the following steps:

- Combining the application of a hydrologic model as learning tool and forward operator with global sensitivity analysis;

- Selections and comparison of physiographic and dynamic response fingerprints within the target catchment along with exploration of possible linkages;

- Comparison of simulated and observed dynamic fingerprints as respective constraints;

- Check and assess consistency of parameter sets to reproduce several fingerprints. Evaluate the most consistent ones against observed streamflow time series;

- Evaluation of dependent parameter sensitivities with respect to streamflow and the selected fingerprints;

In the following we give further details of the model mHM, the FAST method and our selected similarity indices.

## 3.2 The mesoscale Hydrologic Model (mHM)

The mesoscale Hydrologic Model, conceptualised on the basis of grid cells applied to a wide range of mesoscale river catchments $(10^1 - 10^4 km^2)$ (Cuntz et al., 2015; Samaniego et al., 2010a, b, 2011; Kumar et al., 2010) accounts for diverse processes of the hydrologic cycle: Canopy interception, evapotranspiration, snow, soil moisture dynamics, infiltration and overland flow,



interflow, subsurface storage and groundwater recharge, baseflow, discharge attenuation as well as flood routing (Samaniego et al., 2010b; Kumar et al., 2010). Three model levels of gridded information are implemented in mHM (for more details, see Kumar et al. (2010)): morphology, hydrology, meteorology, with $l_0 \ll l_1 \leq l_2$ denoting the relative sizes of the grid cells at the respective data level. In this study both $l_1$ and $l_2$ were set to a spatial resolution of 1000 m, whereas for level-0 with

the morphological catchment characteristics, a finer resolution of $l_0 = 200$ m was selected as an adequate spatial discretisation at this hydrologic scale. The mHM parameterisation is substantially based upon a simultaneous regionalisation technique (see also Hundecha and Bárdossy (2004); Götzinger and Bárdossy (2007)) called multiscale parameter regionalisation (MPR) (Samaniego et al., 2010b) to account for the physiographic sub-grid and hydrologic process variability. Level-1 hydrologic process parameters are derived from (pedo-)transfer functions with coefficients (in the following also referred to as (global)

mHM parameters) related to physiographic characteristics at level-0. Hence, mHM is calibrated indirectly, altering the level-0 parameters instead of hydrologic level-1 parameters. This procedure not only reduces the problem of overparameterisation (52 global parameters) or the dependence on specific hydrologic scales (Beven, 2001) but also diminishes the amount of time to be spent on grid-wise calibration.

## 3.3    Dependent parameter sensitivity

### 3.3.1    The Fourier Amplitude Sensitivity Test (FAST)

The Fourier Amplitude Sensitivity Test (FAST) introduced and tested by Cukier et al. (1973, 1975); Schaibly and Shuler (1973) is a partial variance-based method to determine global sensitivities of parameter changes on the outcome of monotonic and non-monotonic numeric models (Frey and Patil, 2002; Song et al., 2013) with a limited number of evaluation runs. For a detailed explanation of the FAST method the reader might refer to Reusser et al. (2011).

The method was computationally enhanced to extended FAST (eFAST) by Saltelli et al. (1999) merging the advantages of FAST (high efficiency) and Sobol's technique (high order interaction terms and total sensitivity effects) for global sensitivity analysis (Sobol, 1993, 2001). FAST was originally applied to study parametric model sensitivities of chemical reaction systems. The method has since been used in several fields such as atmospheric (Rodríguez-Camino and Avissar, 1998), ecologic

forestry (Song et al., 2013), hydrogeologic (Fontaine et al., 1992) or geologic nuclear waste disposal modelling (Lu and Mohanty, 2001). More recently, a few studies also include the application of (e)FAST to hydrologic modelling (Guse et al., 2014; Reusser et al., 2011; Reusser and Zehe, 2011; Zajac, 2010). Sanadhya et al. (2013) employed eFAST in combination with the Soil Water Assessment Tool (SWAT) (Arnold et al., 1998) to analyse critical interactions of snow-related parameters in mountainous watersheds.



### 3.3.2 FAST implementation for mHM in the study area

In this study, we implemented the FAST method (Reusser, 2014) to initially conduct a sensitivity analysis on mHM with a focus on six global mHM parameters (see Table 1) explaining a substantial part of the model output variance (see also Cuntz et al. (2015)) in the upper Lenne basin at gauge Bamenohl (BAM). The parameters were pre-selected by local sensitivity analysis in terms of different indices for streamflow dynamics and water balance. Subsequently, a pre-run of FAST including 14 parameters previously identified as most sensitive was performed at BAM. For this study FAST was applied again on a subset of six mHM parameters showing highest sensitivity coefficients in the pre-run.

Table 1 presents both these six mHM parameters and their value ranges determining the number of FAST variations (i.e. mHM model runs). The parameter intervals were basically selected from the mHM literature (Samaniego et al., 2014) and then slightly extended based on results from numeric experiments within local sensitivity analysis. In our case, the FAST method requires altogether 91 model runs (variations) to assign sensitivity coefficients to each of the six mHM parameters (see Fig. 1). Parameter-related sensitivity values have been evaluated for each of the 14 headwater subcatchments (see section 5.4).

All other mHM parameters were fixed on pre-calibrated reference values found via global automatic optimisation using the dynamically dimensioned search (DDS) algorithm (Tolson and Shoemaker, 2007) at gauges BAM and Wenholthausen (WEN). DDS calibration results coming from two different precipitation data sets: 12 precipitation stations with Thiessen polygons (Thiessen, 1911) and HYRAS (Rauthe et al., 2013), for the six study-relevant mHM parameters are also depicted in Fig. 1. Covering the whole of Germany and several adjacent regions the HYRAS data set was originally derived from 6200 rain gauges for the period from 1951 to 2006. With $1\ km^2$ grid size a comparably highly resolved daily precipitation data set was available for pre-calibration and FAST-mHM simulations.

Apart from global sensitivity analysis, FAST can be used to span an ensemble spectrum of streamflow simulations. In this study FAST variations provide an ensemble of 91 different streamflow time series for further analysis of model based hydrodynamic response and subsequent parameter identification (see section 3.4.2 and section 5.4).

Based on the ensemble of 91 streamflow simulations we derived ensembles of 91 realisations for dynamic fingerprints characterising different streamflow attributes, to assess their dependent parameter sensitivity as elaborated in the next section.

### 3.4 Fingerprints for regional parameter estimation

### 3.4.1 Relevant physiographic characteristics

On the basis of the available physiographic and meteorologic data a set of 12 indices characterising similarity of the climate and the physiographic setting (see Table 2 and Fig. 2) was selected to



1. capture characteristics that control relevant hydrologic functions of a mesoscale catchment as defined e.g. by Black (1997) such as partition, storage and release of water,

2. relate physiographic and climatic characteristics to sensitivity-confined hydrodynamic response fingerprints,

3. keep the option open for additional research to perform cluster analysis for catchment similarity classification and,

4. to further investigate and avoid the hydrologic similarity paradox described in section 1.3.

Each of these fingerprints were assigned to one of five classes proposed by Yadav et al. (2007) representing the physiographic functions in the panoply of hydrologic processes. The choice of physiographic fingerprints originates from correlation analysis of catchment descriptors within each category (Yadav et al., 2007), multivariate statistical analysis techniques (Di Prinzio et al., 2011), from practical experience with slope-averaging measures in regionalisation models (Plate et al., 1988) and from comparison of methods for baseflow separation (Duband et al., 1993; Eckhardt, 2008, 2012). For subsequent research on the topic of classification using non-redundant hydrologic similarity indices, an alternative composition of physical fingerprints might certainly be conceivable. Ley et al. (2011) for example based a classification study on a set of six physical catchment characteristics including a fingerprint characterising zones of significant groundwater recharge. Likewise, additional similarity indices coping with anthropogenically altered hydrologic settings might be considered in the context of future socio-hydrologic studies.

### 3.4.2 Selected fingerprints for streamflow generation

Six hydrologic fingerprints reflecting the integral hydrodynamic response characteristics of the mesoscale headwaters were derived from streamflow and precipitation data (see Table 2).

One dynamic fingerprint was chosen for this study, each characterising streamflow generation during high (High Flow Discharge), average (Runoff Ratio) and low (Baseflow Index) flow conditions, the variability of streamflow (Coefficient of Variation), the frequency of flow events (High Pulse Count) as well as the rate streamflow changes in a catchment (Slope of Flow Duration Curve). This subset of hydrologic fingerprints is regarded as a reasonable prelude open for later expansion to further response signatures. The fingerprints were selected and classified in accordance with Olden and Poff (2003) and Yadav et al. (2007) who took the study of redundancies via linear and Spearman rank correlation (Yadav et al., 2007) and multivariate PCA (Olden and Poff, 2003) of indices as a basis for their choice of fingerprints.

The response indices were simultaneously determined from the observed and from the simulated ensemble of 91 streamflow time series originating from FAST-mHM parameter variations (see Fig. 2 and section 3.3.2). In addition to the physiographic and climatic characteristics of headwaters, the dynamic response fingerprints were used as a means to constrain the FAST parameter space keeping the observed quantities as a benchmark in the first instance. This provides an insight into the model performance based on FAST ensembles, i.e. multiple mHM parameter sets.





Whether weak or strong, constraints have to be thoughtfully considered, since too strong constraints can result in an overconditioning of the parameter space, leading to a situation with non-resolvable constraints (Zhang et al., 2008) which can highly limit the predictive power of the model causing a complete rejection of all parameter sets. If confining limits are chosen that are too weak the identification of behavioural model set-ups will be complicated and equifinality of retained models (Beven, 2006) might be introduced. If the performance of the streamflow ensemble simulation is rated as very poor, i.e. the distance to the observation is large, a re-run of FAST might be helpful in the search for behavioural parameter sets. The closest simulation results in terms of the underlying observed response fingerprints could be taken to determine the parameter range needed for revised sensitivity analysis.

A special focus was set on the observed flow duration curves (FDCs) of the 14 catchments serving as constraints for the parameter space of the ensemble simulation (see Fig. 3).

Slope values between 33 % and 66 % flow exceedence were derived as fingerprints that represent the rate streamflow changes for intermediate flow events. The consideration of FDCs as are commonly employed to indicate and classify catchment response functioning (Yilmaz et al., 2008) enables analysis of observed and simulated streamflow data - free of autocorrelation and time dependency e.g. in contrast to classical hydrograph inspection. A pronounced sensitivity in terms of the vertical redistribution of soil moisture was further recognised by Yilmaz et al. (2008). Sequential calibration can be performed on curve sections of special importance in terms of catchment response for flow segments of high, medium and low magnitude. Westerberg et al. (2011) showed the advantages to use FDCs as an objective function for model calibration with behavioural parameter sets setting up an uncertainty framework based on a limit of acceptability GLUE approach (see also Blazkova and Beven (2009)) for FDC calibration on selected evaluation points (EPs) of the FDCs. Singh et al. (2014) point out that depending on physiographic similarity classification of 83 gauged US catchments with rather few anthropogenic influences on the hydrologic regime, the slope of the FDCs in addition with a few other streamflow response characteristics (e.g. baseflow index and runoff ratio) can constitute a useful constraint for parameter transfer to ungauged sites. Yu and Yang (2000) employed a methodology whereby synthetic flow duration curves of ungauged Taiwanese sites were derived in order to use them as an objective function preconditioning the parameter space of the applied HBV model.

## 4 Study area

### 4.1 The Ruhr catchment - An anthropogenically influenced system

The Ruhr, with a catchment area of 4.485 $km^2$, originates from a spring about 670 m a.s.l. on the northern slope of the Ruhrkopf in the Hochsauerland (about 842 m a.s.l.) and joins the Rhine at Duisburg-Ruhrort (about 20 m a.s.l.) after 219 km. The landscape characteristics of the Ruhr catchment range from densely wooded and scarcely populated lower mountain ranges in the Sauerland to widely impervious urban areas in the river valleys and in the western part close to the mouth. The areas can be attributed to the geology and geography of the Rhenish Slate Mountains to the east of the Rhine (Brudy-Zippelius, 2003). The



average discharge at its mouth joining the Rhine is about 80.5 $m^3 s^{-1}$ (Bode et al., 2003).

With a total of 8 dams and 5 reservoirs the Ruhr with its tributaries forms a very complex hydrologic system. The total stored water surface area of just under 35 $km^2$ equates to about 480 million $m^3$ as the volume of all the water retained behind

the damming structures (Ruhrverband, 2011). An intensive use of water resources (e.g. reservoirs, barrages, sewage plants, withdrawals, inlets, etc.) supplies almost 5 million people with drinking and processing water along the Ruhr and within its catchment area and because of this intensive use, a large amount of water is lost completely (310 million $m^3$) (Bode et al., 2003). Annual evaporation of about 35 million $m^3$ can be estimated for the known inundated areas by using a simple empirical method e.g. by Haude (1955). The large anthropogenic influence on the hydrologic cycle, particularly water export, low wa-

ter elevation and evaporation from the reservoirs, causes significant discrepancies and an overestimation of the water balance when applying conceptual hydrologic models to river reaches downstream of the reservoirs e.g. at gauge Villigst (see Fig. 4). Employing model efficiency measures without taking account of anthropogenic changes in the water balance due to damming and water management (Morgenschweis et al., 2003) is highly inappropriate for estimating the quality of a model in the Ruhr catchment. Comparisons with the clearly affected observed time series generally rule out high model efficiencies as the model

reproduces the unaffected hydrologic system without the use of water resources.

## 4.2 Ruhr headwater catchments

The present study concentrates on 14 headwater subcatchments of the river Ruhr and upper reaches of its tributaries (e.g. Bigge, Lenne and Möhne) where the assumption of unaffected hydrologic regimes is justified to a much higher extent (Fig. 4). The

pre-selection of headwaters is an essential factor in avoiding additional uncertainty in model parameterisation (Liu and Gupta, 2007; Fenicia et al., 2008).

The headwaters are situated in the eastern rural part of the Ruhr basin with higher altitudes, covering an area of 1989.2 $km^2$ in total, where individual catchment sizes range from 28.7 $km^2$ at gauge Amecke (AME) to 453.1 $km^2$ at gauge Bamenohl

(BAM). Average catchment slopes vary between 10.8 % (Rüblinghausen) and 26.1 % (Kickenbach). The dominant form of land cover is forest (39.7 % - 87.3 %) followed by pasture (0.8 % - 47.5 %), cropland (7.6 % - 43.9 %) plus a few predominantly dispersed settlements (0.0 % - 13.2 %) with a subordinate role with regard to the integral hydrologic response characteristics (see Fig. 2a). The climatic conditions are humid-warm-temperate (Göppert et al., 1998) with warm summers and moderate winters, with annual mean temperature values between 8.45 °C and 5.45 °C from lower to higher altitudes in the eastern

part of the headwater catchments. Annual precipitation ranges from 959 mm in the northern to 1473 mm in the south-western catchments (Ruhrverband, 2013).





## 5   Results

### 5.1   Constraints on parameter space

As described above in section 3.4.2 the aggregated dynamic response of headwater catchments has been derived in the form of six streamflow indices considered as regionally typical hydrologic fingerprints. Fingerprints were evaluated for observed and
simulated time series. The former act as constraining quantities for the latter where an ensemble of model runs was made available from 91 different parameter combinations spanning the FAST-mHM parameter space. As well as box plotted catchment physiographic and climate characteristics, Fig. 2b shows the results in an overview for all six dynamic fingerprints considered in the form of the simulated ensemble and of observed values at the 14 gauges. Figure 2 also includes the baseflow index (BFI) as an intermediate hydrologic catchment signature between a physiographic descriptor and a dynamic response characteristic.
For the next step, the observed and simulated values of two response fingerprints were plotted as scatter points against each other. Figure 5a-d shows the values of observations and ensemble simulations for four pairs of dynamic fingerprints at seven selected gauges. The observed quantities are plotted as circles where the centre represents the actual 2-d combined fingerprint value and the radius serves as a qualitative constraint on the ensemble of Fast-mHM simulations. For each radius a scaling factor is introduced which can be regarded as a measure of distance orientated towards the overall data spread of the 14 derived
values of one fingerprint. The factor is determined as a multiple $n$ of the average of the two standard deviations $\sigma$ of the pairs of fingerprints (FP) considered in Fig. 5a-d.

Simulated value pairs which plot inside the circles can be rated as behavioural, assuming the hydrologic response similarity of the model and the hydrologic system in terms of the two combined response indices (Blöschl et al., 2013). Thus, the under-
lying parameter set can be shortlisted potentially together with other behavioural parameter combinations. Likewise, Euclidean or Chebychev distance values between the 2-d combined observed and each of the simulated plots might serve as feasible measures of functional performance to evaluate FAST-mHM ensembles (see section 5.3).

Consider the case where the coefficient of variation ($CV$) as a hydrologic fingerprint of longterm streamflow variability is
combined with the slope of the respective flow duration curve (SLFDC) indicating the rate of change in streamflow of average magnitudes (33 % - 66 % exceedance) (see Fig. 5c). The first point to recognise is that for each catchment the simulated FAST-mHM ensemble clusters in an arrow-like sample space showing relatively small scatter along the two branches. The head is positioned at the right-hand end of the horizontal axis with larger values of SLFDC of about 2.1 % vertically central at a CV-value of 1.4 connecting the two branches of the arrow. The lower branch where the simulation produced positively
correlated scatter between both fingerprints is regarded as the behavioural subset of the model runs. The observed $CV$ values cluster in a comparably dense range of 1.1 to 1.3 in the right half of the lower branch of the arrow head implying similar hydrologic functioning for this response signature of streamflow variability.





Most of the circles contain at least a couple of corresponding simulated data points. However, in the case of Fig. 5c ($CV$ against SLFDC) the number of behavioural model runs varies among the seven catchments, which portends that the constraining effect by a uniform distance measure (circle radius) diversifies catchment specifically. Gauge Meschede (MES) exhibits only one model run plotting at the low central edge of the circle (medium brown colour), still underestimating streamflow variability of the relatively large catchment with the highest elevation range, a steep catchment slope and a lower weighted slope along the main flow path, all at the same time. Fast reacting interflow components capture a more substantial part of the hydrograph which can be verified by the baseflow index based on the observed data that shows the lowest value (BFI=0.3) among all 14 catchments (see Fig. 2a). In other more frequent cases such as gauge Kickenbach (KIC) (grey) several parameter sets produce results close to the observed reference values (circle centre). Similar results were obtained from the combination of High Pulse Count (HPC) with SLFDC plotted in Fig. 5b. Circles can clearly constrain the bulk of model simulations with varying strength.

The combined scatter of $CV$ against HPC shows a kind of expected image of correlated fingerprints (Fig. 5d). Streamflow variability and the yearly frequency of flow events above three times the average streamflow represented by $CV$ and HPC exhibit a strong positive correlation with $\rho = 0.84$ for the observed fingerprints at the 14 headwater gauges. The ensemble of model runs even indicates a higher strength of linear relationship of the two fingerprint variables ($\rho = 0.96$). This clear linear correlation can be rated as a sign of redundancy of two fingerprints describing similar response characteristics. Obviously, the seven circles in Fig. 5d act as constraints for the scatter of simulations wherein all observations are centrally positioned, indicating intermediate streamflow dynamics.

In exactly the same manner dynamic response fingerprints were combined with the intermediate fingerprint BFI (introduced in Table 2 and Fig. 2) both for observed values and for the model simulated ensemble output. The pairwise analysis consistently including BFI shows somewhat similar findings as illustrated and presented here. Likewise, for example the contribution of SLFDC leads to an arrow-like simulated sample space and combinations with $CV$ exhibit a linear correlation to a certain degree that here is distinctly stronger for the simulated ensemble ($\rho = 0.91$) than for the observed data ($0.52$). The coupling of BFI with HPC, different from the results in Fig. 5b, though less marked, also shows this correlation ($\rho = 0.87$; $0.62$). Similar site specifically varying constraining effects on the ensemble by the observed combined fingerprints were found in this analysis based on BFI.

## 5.2 Linkages of physiographic characteristics and hydrodynamic response

Different from those itemised in section 5.1, fingerprints that characterise the hydrologic response of a catchment were only plotted on the ordinate, whereas both physiographic and climate catchment charactericts were plotted on the abscissa. In Fig. 6a-d SLFDC representing the rate of change in average streamflow was combined with four physiographic attributes forest land cover (FOR), average catchment slope (SLOPE), longest drainage path (LDP) and the flow accumulation (FACC)



describing land use, topography and landform, respectively.

For most of the gauges (11) more than half of the ensemble of 91 simulations shows an underestimation of the observed SLFDC values which vary only slightly from 1.3 % at Meschede (MES) to 1.9 % upstream the Möhne reservoir in Neuhaus (MOE). Nevertheless, differences in SLFDC, though weakly pronounced, indicate a diverse composition of hydrologically relevant structural elements. Trying to explain in this context certain salience at selected headwaters, the four highest SLFDC values were found at MOE, HER, KIC and BAM (Fig. 6). These are the gauges exhibiting both highest percentage of forest land cover (FOR) (66.8 % - 87.3 %) (Fig. 6a) and, with the exception of MOE, steepest average catchment slopes (21.1 % - 26.1 %) (Fig. 6b). Other dynamic response fingerprints such as HPC and HFD (see Fig. 2b) plot at values close to the overall average of all headwaters which portray jointly a moderate hydrologic response for these four catchments in terms of high flow conditions.

### 5.3 Consistency and performance of behavioural parameter sets

In this subsection behavioural FAST-mHM parameter sets (see also section 5.1) were analysed and evaluated against observed streamflow time series. The evaluation of behavioural parameter sets was done with respect to their consistency in reproducing one fingerprint of hydrodynamic response, and combinations of two, both at different locations with slightly different physiographic settings and for other signatures, which reflect further major response characteristics of the mesoscale headwaters.

In Fig. 7a the Nash-Sutcliffe model efficiency (NSE) criterion (Nash and Sutcliffe, 1970) is plotted against the relative error between simulation and observation. In this case, evaluation is shown for the best performing parameter sets, in terms of each dynamic fingerprint at seven selected headwater gauges. Different colours were chosen to mark the headwaters, whereas the distinguishable form types of the markers indicate individual fingerprints. Each marker is labelled by a number signifying the parameter set in the ensemble found to perform most behaviourally in the reproduction of the fingerprint value. Additionally, Fig. 7b illustrates a subset of Fig. 7a condensed to the two fingerprints Runoff Ratio (RR) and the slope of the flow duration curve (SLFDC) at two gauges Amecke (AME) and WEN, extended by the results for the whole of the simulated ensemble of FAST-mHM parameter sets (model runs).

It can be frequently observed, that single dynamic response fingerprints are most behaviourally reproduced by fingerprint specific parameter sets. Furthermore, in this case, the fingerprint specific parameter sets seem to be unique for distinct headwater gauges, fairly closely located in some cases. Figure 7b also shows that for the simulation of RR the parameter sets with smallest relative errors plot close to the optimal solution within the ensemble concerning NSE, while both best SLFDC results show negative NSE values and are way out, especially at the gauge AME.

Contrarily to the findings for one single fingerprint, the reproduction of two combined dynamic fingerprints (section 5.1) can more frequently result in a consistency of most behavioural parameter sets plotting most closely to the observed pair of





fingerprints. The scatter plots in Fig. 7c and Fig. 7d evaluate the simulated ensemble for combined fingerprints (see also Fig. 5, section 5.1). NSE values are plotted against the Euclidean distances ($D_E$) between four pairs of observed and corresponding simulated fingerprints. Parameter sets with smallest $D_E$, in the following are shortlisted as the most behavioural for the reproduction of combined fingerprints, and are labelled and markers are bold-framed. Figure 7c illustrates a fairly consistent picture

whereby parameter set 78 appears to be the best fit to the combination of SLFDC and HPC at both gauges and at least for WEN in terms of the pair HPC-CV. Figure 7d only partly conforms to this image of consistency, revealing solely set 78 as the most behavioural for RR combined with SLFDC at gauge AME.

Up to this point the evaluation of model performance via FAST-mHM ensembles with observed values indicates that most

behavioural parameter sets, which best reproduce single or combined dynamic fingerprints, might not necessarily lead to best results among all ensemble members in terms of the NSE performance measure. This assumption is supported by Fig. 7e-f, where the overall five most behavioural parameter sets for combined fingerprints (referred to the seven shown headwaters), the Spearman's rank correlation coefficient $\rho_s$ (e) and the water balance percentage error (WBE) (f) are scatter plotted against NSE within the ensemble.

Sets identified as behavioural combinations of six mHM parameters were further analysed in terms of their distances among themselves in the 6-d vector space. Predominantly, we used the Chebychev distance to detect the placement of a specific set within the entire FAST-mHM ensemble. Chebychev distances, along with the corresponding Euclidean values in reference to behavioural set 78 for combined dynamic fingerprints, were incorporated for all parameter sets. Generally, Chebychev

distances (also Euclidean) of five best parameter sets (of combined fingerprints) vary relatively with the chosen reference parameter set. Though placed very closely to 78, parameter set 64 does not necessarily show the most similar performance values (see Fig. 7e-f). This finding might be partly triggered by the nature of the maximum norm underlying the Chebychev distance metric emphasising the largest differences in higher dimensioned parameter values without necessarily having the largest influence (sensitivity) on runoff generation. Generally, Chebychev distances vary site-specifically as well as among the

different fingerprints. In our case, distances are generally found to be smaller between the parameter sets which best reproduce combined fingerprints (18.7) than between site and fingerprint specific sets that reproduce a single fingerprint value most adequately (23.5 (WEN), 25.8 (AME)).

Since simulations based on behavioural parameter sets frequently showed rather poor results, if performance measures like

NSE and $\rho_s$ were considered (Fig. 7), visual inspection of time series was conducted. This allows better to judge why these sets do not perform well during simulation, whether e.g. peaks are to small or timing errors possibly affect the results. As illustrated in Fig. 8a and 8b streamflow hydrographs that originate from the most behavioural parameter sets in terms of the reproduction of two combined dynamic response fingerprints (SLFDC-HPC, HPC-CV, RR-SLFDC and SLFDC-CV, see also Fig. 5 and Fig. 7) were further analysed and compared to the hydrographs derived from observational data as well as to the





simulation revealing highest Nash-Sutcliffe model efficiency values within the FAST-mHM ensemble.

Both examples in Fig. 8 show that two hydrographs plot at the lower, under- (SLFDC-CV) and upper, overestimating (SLFDC-HPC) end of the ensemble spectrum during average and low flow periods. Streamflow spectrums are simulated with

high dynamics (SLFDC-CV) or high damping (SLFDC-HPC), leading to an over- (SLFDC-CV) or underestimation (SLFDC-HPC) of observed high flow conditions. The combination of RR-SLFDC exhibits intermediate (Fig. 8a) or similar highly damped (Fig. 8b) streamflow characteristics, whereas parameter sets reproducing HPC-CV most adequately consistently result in highly damped hydrographs with low dynamics at both gauges (Fig. 8a and Fig. 8b). Surprisingly the damping sets 78 and 85 produce contrary results in terms of the WBE at the two gauges (Fig. 7f). The model runs based on the two parameter sets

78 and 85 show water balance percentage errors of very low 0.95 % and 3.49 % at WEN and high 48.1 % and 52.9 % at AME.

Parameter set 60 in the ensemble of 91 FAST-mHM sets frequently produces highest NSE model efficiency values and reaches 0.75 at WEN and 0.64 at AME, respectively. Generally, NSE values do not exceed the 0.77 mark at all headwater gauges, constituting an acceptable adjustment, if kept in mind that the six considered parameters were not calibrated but varied

according to FAST sensitivity analysis. Nevertheless, certain inadequacies can be observed, especially high flow peaks are commonly not properly matched, low flow recessions are underestimated or model timing errors can be detected (Fig. 8a and Fig. 8b), preventing higher efficiencies, likewise for the results based on behavioural parameter sets.

## 5.4    Parameter identification via sensitivities

### 5.4.1    Sensitivity reference

The application of parameter sensitivities as a means to identify and focus more specifically on the most influential subset of the parameter space constitutes the third column of the approach. Global sensitivity analysis with FAST provides sensitivity index values (see Table 1 for sensitivity ranges) for the six analysed mHM parameters and 14 considered headwaters. The sum over all six sensitivity values per timestep reaches a maximum value of 0.43 at HER and can, in principal, maximally equal 1.0

for the entire parameter space (Cukier et al., 1975; Reusser et al., 2011), i.e. for the whole of 52 global mHM parameters in this case.

For every timestep the observed streamflow value can be related to sensitivity values assigned to each model parameter. Time dependent graphs are depicted in Fig. 9a and Fig. 9b showing sensitivity dynamics for the six mHM parameters at the gauges

WEN and HUE in the hydrologic year 2009/10. WEN and HUE were selected as representatives of two major groups showing essentially distinct sensitivity properties of parameters. From a preliminary result of visual inspection and in accordance with the findings in section 5.3, the model performance is obviously higher for the larger WEN catchment and in an acceptable range, in contrast to the poor ensemble simulation results at HUE shown in Fig. 9c. For that reason and for reasons of legibility,





some of the plots presented in this section relating streamflow to sensitivity indices were depicted for both gauges in the case of observed streamflow but for the ensemble simulations only for gauge WEN.

Sensitivity hydrographs plotted as solid lines for WEN in Fig. 9a-b indicate some kind of reverse patterns between some
of the six parameters, while AspectcorrPET, representative of others, and correcting potential evapotranspiration data aspect specifically, shows an opposite sensitivity behaviour to RechargeCoeff, determining the percolation rate to the groundwater reservoir. The former one allows the inclusion of the exposition of slopes, controlling insulation, into evapotranspiration estimations. It essentially influences the water balance - and thus streamflow behaviour -, whereby, as expected, highest sensitivity values are reached in summer periods. Conversely, RechargeCoeff exhibits its largest influence on streamflow generation dur-
ing winter, with negligible evapotranspiration rates and substantial amounts of groundwater recharge in this region.

In the next instance, at each modelled timestep, FAST parameter sensitivity indices were ranked in descending order to obtain parameters with the highest influence on the generation of streamflow for every simulated day. The most sensitive parameter at each timestep was then related to observed streamflow hydrographs plotted in Fig. 10 together with the FAST-mHM ensem-
ble. The hydrograph for WEN is mainly dominated by the colours of the two opposing parameters AspectcorrPET (blue) and RechargeCoeff (orange). Indeed, shortly recessed by soil moisture parameters ThetaSDb (light green), OMcontforest (green) or Ksconst (light blue), both in the summer period of 2010 and on the limbs of flow recessions, AspectcorrPET, controlling evapotranspiration, has the highest influence on simulated streamflow generation. During the winter season RechargeCoeff reveals the longest periods of highest sensitivity values where percolation and groundwater recharge dominates.

At streamflow gauge HUE essentially distinct sensitivity properties of parameters can be observed. Hydrograph sequences of the highest parameter sensitivities are only attained by Ksconst, AspectcorrPET and especially ThetaSconst, a global soil moisture pedotransfer-parameter used to estimate water content at soil saturation in order to calculate the soil water content via the equation proposed by van Genuchten (1980) and discussed by Zacharias and Wessolek (2007), seem to strongly influence
streamflow dynamics at this site, interacting inversely with Ksconst (Fig. 9 and Fig. 10).

Parameter sensitivities were also related to FDCs to sequentially evaluate portions of the streamflow spectrum and to identify process-relevant parameters at different flow magnitudes. Four sequences with different intervals of streamflow exceedance for the observed (WEN and HUE) and simulated ensemble (WEN) of FDCs are pictured in Fig. 11a-d. Observed FDCs, provided
with parameter information of highest sensitivity at every point of flow, give an insight into differences of dependent parameter sensitivity for the whole spectrum of streamflow, revealing regional differences in both model and catchment functioning at distinct headwaters.

For WEN, RechargeCoeff clearly dominates the first five percent of streamflow exceedance with highest flow values (Fig. 11a),
whereas the evapotranspiration parameter AspectcorrPET is most influential on runoff generation in the lowest 5 percent of





flow (Fig. 11d). In the large mid section of the FDC for WEN (Fig. 11b and Fig. 11c) a rather heterogeneous composition and permanent change in the highest parameter sensitivity index values becomes obvious. In the headwater catchment of HUE the soil moisture parameter ThetaSconst predominates in high and low flow sequences as well as partly so in the mid sections, mostly alternating with Ksconst, another soil moisture regression parameter, to estimate the saturated vertical hydraulic con-

ductivity (Cosby et al., 1984) within field capacity calculations.

Although some of the parameters named above frequently show the highest values of partial parameter sensitivity, the influence on model-generated streamflow is certainly a complex interplay of different sensitive parameters, which act as a team and conceptually reflect the diverse composition of physiographic and climatic factors affecting the natural hydrologic func-

tioning of catchments at various scales. Nevertheless, the reference to parameters with the highest partial sensitivity provides a feasible means to set the focus more precisely on calibration parameters having essential influences on streamflow generation.

### 5.4.2    Sensitivity duration

Besides the reference of highest parameter sensitivity values to streamflow hydrographs and FDCs we also employed Sensitiv-

ity Duration Curves (SDCs) to be able to further specify the parameter identification process. Analogous to FDCs, a SDC is a cumulative frequency curve that each applies to one parameter, to one (type of) catchment and to the period in which sensitivity analysis is performed. SDCs were developed by arranging the assigned daily FAST sensitivity index values according to their magnitude in ascending order and by plotting as a line against the percentage of time during which the sensitivity equalled or exceeded the specified values. The implementation of SDCs helps both to identify varying functions of parameters more easily

and to better concentrate on parameters with higher sensitivity duration, exhibiting more substantial influences on simulation results.

SDCs are plotted in Fig. 12a-d for four of six analysed mHM parameters at each of the 14 headwater gauges covering the simulation period from 2003 to 2011. OMcontforest, which describes the portion of organic matter (OM) in soil to account

for effects of organic matter on soil water retention characteristics in forested areas (Zacharias and Wessolek, 2007), shows a similar sensitivity behaviour in all 14 headwaters (Fig. 12a). The highest sensitivities are reached during intermediate streamflow conditions at rising and falling limbs (see also Fig. 9 and Fig. 10). A maximum sensitivity value of 0.176 is reached in the headwater catchment of MOE with the highest percentage of forest cover (87.3 %) (Fig. 2a), which indicates the direct connection of the parameter to forest as a significant land cover type in this region. Nevertheless, large periods exist when streamflow

generation is not crucially influenced by this predictor, in our case with soils showing low OM: Zacharias and Wessolek (2007) concluded likewise, even proposing to renounce OM as a input variable, marginally affecting simulation output.

AspectcorrPET (Fig. 12b) shows similar progressions of sensitivity values at all gauges within the lower 50 %. Conversely, from this point on and for the upper half of the curves, a spread in SDCs can be recognised. For some of the headwaters the




slopes of the SDC decrease and the highest sensitivities do not exceed 0.08, whereas the rest of the SDCs keep slopes constant beyond the split point and reach, with a maximum of 0.132 for WEN, generally higher magnitudes of sensitivity index values. Especially in the south-western headwaters (e.g. BOE or HUE) with lower relief energy, the aspect as a physiographic predictor seems to exhibit smaller influences on evapotranpiration and water balance than in catchments showing a bold relief, with valleys more deeply incised where the factor of exposure dominates evapotranspiration processes.

This split picture is also confirmed by Fig. 12c and Fig. 12d where SDCs are plotted for ThetaSconst and RechargeCoeff. For the south-western catchments, including the largest headwater BAM where a more intermediate sensitivity behaviour shows up, ThetaSconst (Fig. 12c) has an increased influence in large parts of the spectrum, while RechargeCoeff (Fig. 12d) seems to be steadily more sensitive in the eastern and northern headwaters. For the reproduction of observed streamflow, soil water retention characteristics along with runoff generation from the unsaturated zone, seem to play a major role in the former rather than in the latter headwaters where the percolation into the deeper saturated zone and subsequent baseflow prevail in the interplay of hydrologic processes.

# 6 Discussion and conclusions

The main objective of this study was to approach three embedded research questions, whether (1) fingerprints of streamflow response characteristics can act as constraints on the parameter space of a distributed hydrologic model, (2) regional differences in parameter sensitivity can serve as an indicator to learn about spatial differences in catchment functioning and (3) differences in hydrologic catchment behaviour and parameter sensitivities can be applied as explanatory variables for physiographic or climate attributes.

(1) The approach based on six streamflow fingerprints, representing the observed catchment response functioning of 14 Ruhr headwaters, allowed us to effectively constrain the parameter space of a distributed hydrologic model to the most behavioural subset in terms of the specific reproduction of these indices. Observed response fingerprints were found to cluster within typical catchment sample regions. The respective ensemble of simulated counterparts, initially model-specifically grouped, was successfully constrained to the observed sample regions of the hydrologic system. We further found most of the behavioural parameter sets within the constrained subsets to frequently change along with each fingerprint and location. The singularity of parameter sets for fingerprints or location is partly attenuated if two fingerprints are considered as constraint which we defined as a measure of the minimal distance in the two fingerprint sample space. Streamflow hydrographs based both on the combined and fingerprint specific behavioural parameter sets often show particular dynamics with varying or even opposed results if standard efficiency criteria are considered.

These differences might be partially explained by the fact that, contrary to the three chosen measures, dynamic response fingerprints aim at absolute performance values of a specific hydrologic functioning, less normalised than e.g. the NSE, with



higher explanatory value, objectively interpretable, which was also noted e.g. by Schaefli and Gupta (2007). Notwithstanding, results in model performance (NSE, $\rho_s$ and WBE) of parameter sets, outlined in Fig. 7, can be (highly) improved or further corrupted by randomly selecting parameter sets from the group of the five most behavioural ones both within the same catchment (e.g. WEN) or by transferring the selected set to another site (AME).

(2) Region specific differences were not only detected among the dynamic response fingerprints but also for dependent parameter sensitivity by employing FAST on the mesoscale Hydrologic Model. Headwaters can be grouped by sensitivity properties of parameters into two major categories, that also exhibit high and low (five southwestern headwaters) model performances in our example. Parameter effects (sensitivities) on the hydrologic response behaviour can be detected far easier by the option of a reference for streamflow hydrographs or flow duration curves by highest sensitivities. The composition of parameters identified as the most influential for streamflow sequences at different points in time or at variable magnitudes (free of autocorrelation and time dependency) was greater in number and variety in headwaters where the distributed model was found to perform reasonably well in terms of streamflow dynamics and response fingerprints. Sensitivity dynamics were also significantly higher in the eastern headwaters. Sensitivity duration curves allowed us to distinguish differences in parameter sensitivity exceedance probability progress between the major groups. SDCs provided a convenient means for studying varieties of sensitivity characteristics of parameters at sites often revealing crucial hydrologic differences.

These local differences in sensitivity dynamics, variety and duration of parameters might be partly triggered by model independent uncertainties in the spatial distribution of meteorologic forcing data (Song et al., 2013), but can also be interpreted as hint of different hydrologic behaviour caused by factors such as physiographic settings. The RechargeCoeff as a conceptual proxy of catchment structures controlling percolation and groundwater recharge either temporarily exhibits high influences on streamflow generation or solely plays a tangential role which in the southwestern headwaters is compensated for instead by proxies for soil moisture dynamics (Ksconst) or evapotranspirative processes (AspectcorrPET) (Fig. 9a-b and 10). Soil moisture controlled conditions can be associated with the generation of fast reacting or slightly deferred components of streamflow which dominate during high flow periods. Response fingerprints describing the natural characteristics of highflow events (HPC and HFD) and the variability of streamflow (CV) showed highest values in the five southwestern catchments (Fig. 2b). The ranking of highest parameter sensitivities seems to generally capture this special local functioning of flashy response while physiographic characteristics (Fig. 2a) do not intrinsically differ or cluster apart from the eastern group of headwaters.

(3) A selection comprising the most integral catchment attributes or reliable derivatives (Fig. 2a) was set up within catchment functional categories (Table 2) serving as possible structural and climatic similarity metrics for later cluster analysis. Physical descriptors were primarily employed to explore basic linkages to potentially corresponding dynamic response fingerprints connecting observable with predictable hydrologic response variables.





Specifically we found that four highest SLFDC values were found in headwaters (MOE, HER, KIC and BAM), characterised by both highest percentage of forest land cover and, with the exception of MOE, steepest average catchment slopes. Other dynamic response fingerprints plot at values close to the overall average of all headwaters which pictures a jointly moderate hydrologic response character among these four catchments in terms of high flow response functioning. In a qualitative sense

and in this specific case, the vast forest coverage (damping) and marked relief intensity via higher slope averages (stimulating) in the catchment seem to play a certain role for the rate of change in average streamflow events whereas catchment form descriptors such as the longest drainage path (LDP) (Fig. 6c) or the topographic descriptor flow accumulation (FACC) (Fig. 6d) do not seem to have further explanatory value.

Since the nature of hydrologic response is generally composed of and generated by the complex interplay of multiple catchment specific factors (see section 1.3), the explanation of a categoric dynamic fingerprint value at a specific site by one physiographic or climate attribute is crucially hampered. Site and fingerprints specific most behavioural parameter sets further underline the heterogeneity in the interplay of catchment conditions determining hydrologic processes and consequently complicate a reliable model-based mapping of streamflow and its derivatives. As also discussed in section 1.3, conclusions about

hydrologic similarities cannot be easily drawn if the process behaviour represented by dynamic response fingerprints does not conform to causative physiographic catchment descriptors impeding model parameter estimation, not to mention parameter transfer based on feasible regionalisation representable by the given model structure (section 1.3). However, global sensitivity analysis spanning an ensemble of parameter combinations that reproduces streamflow of headwaters in many different ways can give further support in the explanation of similar or distinct response functioning.

This paper has implemented some promising steps toward an analytic framework to (1) investigate hydrologically relevant structural and functional attributes in terms of consistency and feasibility in classifying similar catchments, (2) assess the value of functional constraints for the parameter spaces of distributed hydrologic models, (3) diagnose and explain regional and temporal differences of model dynamics and performance based on global sensitivity analysis. The concept and system-

atic approach thus presented, along with our findings in this study, motivate starting points for additional research and some methodological revision.

(1) A lack of diversity in terms of some of the absolute values of physiographic fingerprints (e.g. wetness index) both unveils similar structural conditions or forcings of proximal locations that helps to sort out redundant catchment characteristics but also

limits the scope within this study area for catchment classification purposes. Contrarily, other physiographic indices such as the forest cover or the average catchment slope were better able to capture important regional characteristics. Scale dependent limitations have to be kept in mind to avoid a levelling out of the explanatory value for a physiographic index (Blöschl and Sivapalan, 1995). Cluster analysis, including mesoscale headwaters with different physiographic and climatic settings, certainly further enables classification by structural similarities of watersheds, which is an imminent development. Some further

fingerprints should replace those found to be redundant in order to capture all hydrologically behavioural components relevant





within the functions of watersheds (Black, 1997) such as the recession behaviour of streamflow or by indices that evaluate timing errors of simulated streamflow (see e.g. Yilmaz et al. (2008)). This step might further clarify potential linkings of functional and structural characteristics and could implicate an additional gain in model performance with regard to hydrograph dynamics and will add explanatory value for sequential calibration on different streamflow magnitudes.

(2) The consideration of combined fingerprint calibration constraints led to a higher degree of consistency in parameter sets which can serve as a feasible trade-off to map physiographic and functional heterogeneity by the given model structural concept. Behavioural parameter vectors can then be classified by their effect on streamflow dynamics within specific physiographic settings with reference to partial sensitivities in the set.

(3) In addition, more parameters maybe analysed within FAST to raise the sum of sensitivity index values per timestep significantly, explaining an even larger majority of variances in streamflow. This step involves the ensemble spectrum to be widened, potentially leading to more behavioural solutions involving the risk for equifinality. Global sensitivity analysis can be directly applied to catchment class-specific dynamic fingerprints, in order to establish feasible relationships of individual

15 (vectors of) parameters and functional attributes.

van Griensven et al. (2006) similarly remarked that local differences indicate that results of global sensitivity analysis for one (group of) catchments cannot be applied directly to other sometimes even closely located places but maybe used as a reasonable estimate within the same catchment category of parameter sensitivity. Therefore, classification by sensitivity duration and

20 dynamics are further conceivable and required as subsequent steps. As correctly recommended by (Song et al., 2013) for a complete analysis of parameter effects on model output, influences due to meteorologic forcings might also be assessed in order to employ the multi-fingerprint approach to ungauged sites or those under substantial anthropogenic influences.

*Acknowledgements.* We acknowledge support by Deutsche Forschungsgemeinschaft and Open Access Publishing Fund of Karlsruhe Institute of Technology.



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





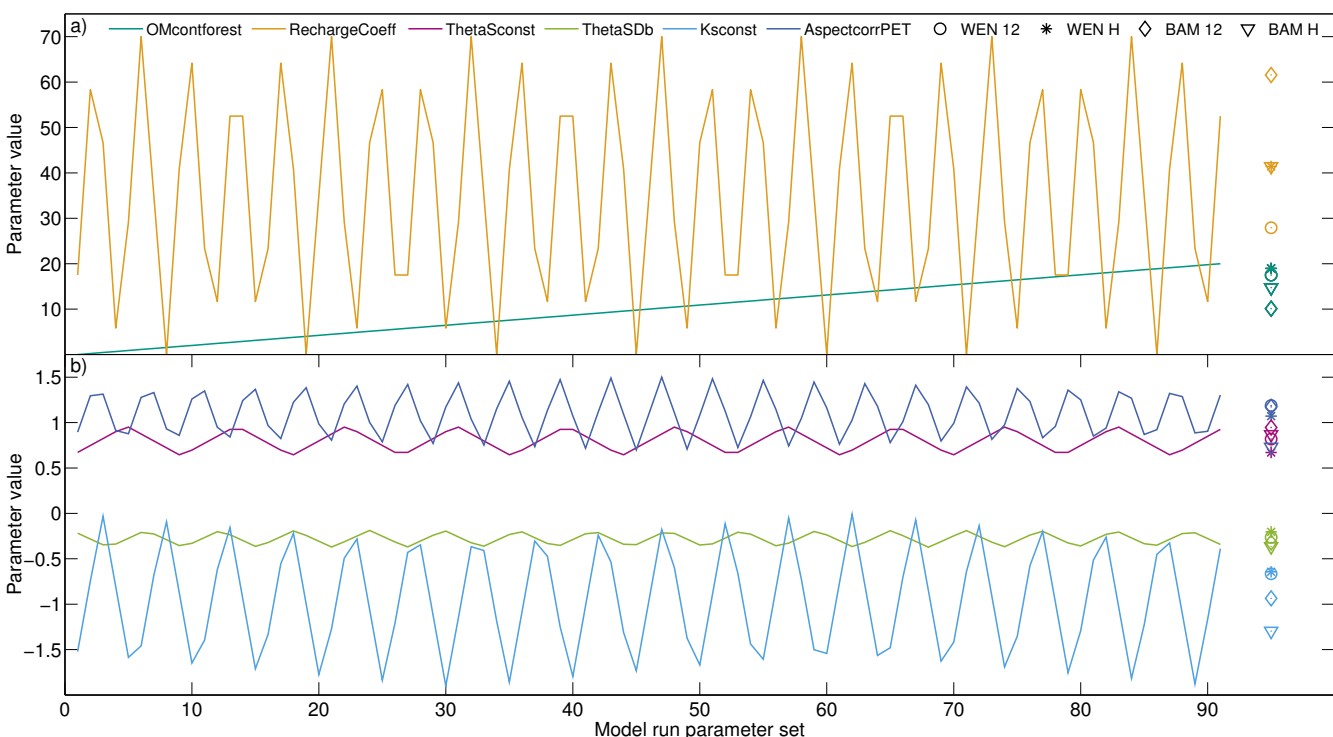

**Figure 1.** 91 FAST variations of the six selected global mHM parameters (see Table 1) plotted as connected curves and mHM parameter values obtained by calibration using the dynamically dimensioned search (DDS) algorithm at gauges Bamenohl (BAM) ($\diamondsuit, \triangledown$) and Wenholthausen (WEN) ($\circ, *$) (see Fig. 4 for gauge locations). DDS calibration was based on two different precipitation inputs: 12 DWD stations with Thiessen polygons (12) and HYRAS gridded precipitation data set (H).





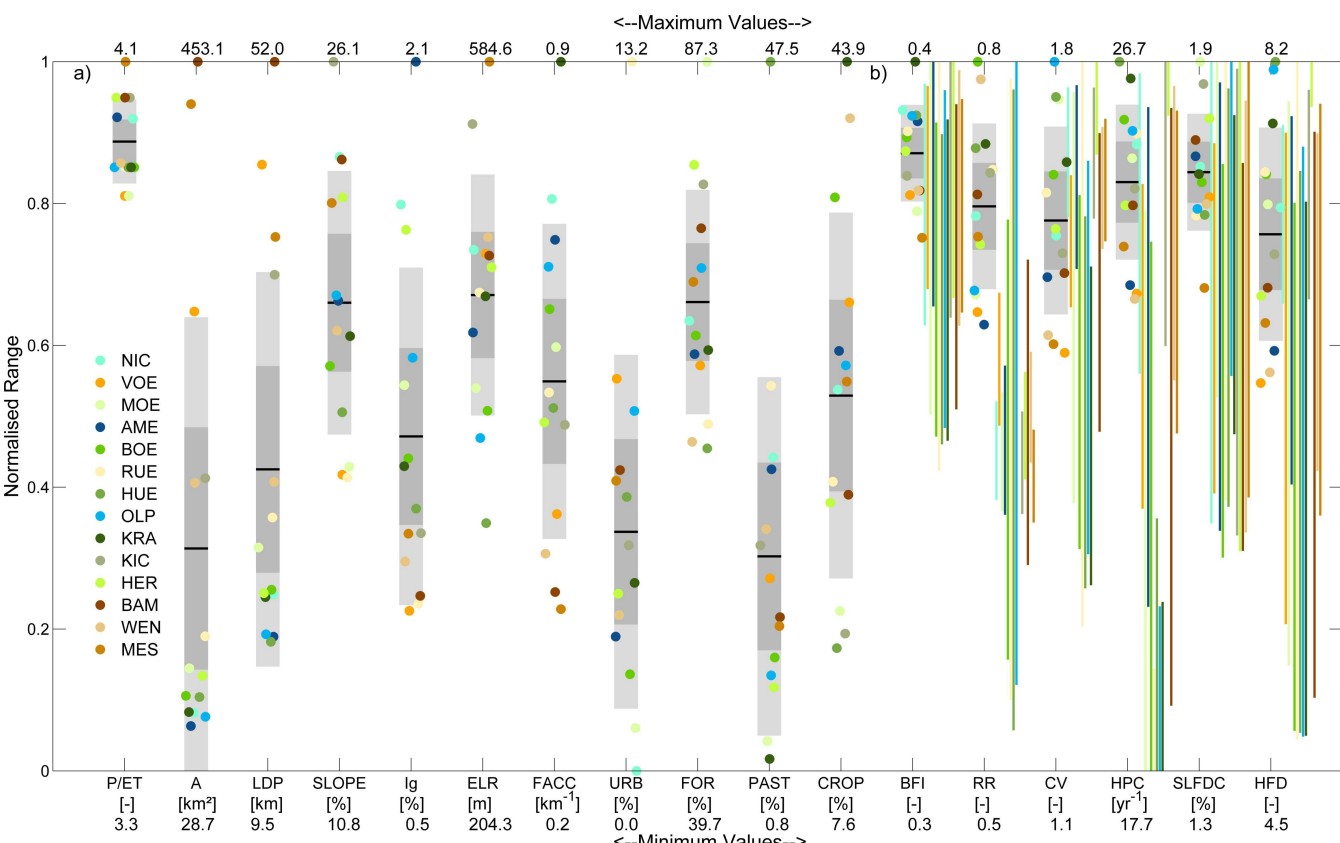

**Figure 2.** Plots of derived hydrologic fingerprints of physiographic and climate characteristics (a) & dynamic response (b) for 14 headwaters (colour coded). Observed fingerprints as shown as dots juxtaposed with corresponding simulated FAST-mHM ensembles depicted as vertical lines in the case of dynamic response fingerprints (b). Observed values superimpose boxes showing appertaining averages (black line), 95 % confidence intervals (grey) & standard deviations $\sigma$ (light grey). Numbers on the lower and upper axis denote minimum and maximum values of fingerprints among the 14 watersheds. In the case of dynamic response fingerprints the extreme values are denoted for indices derived from observed streamflow data (b). Plots are normalised by the respective maximal values among the 14 headwaters.





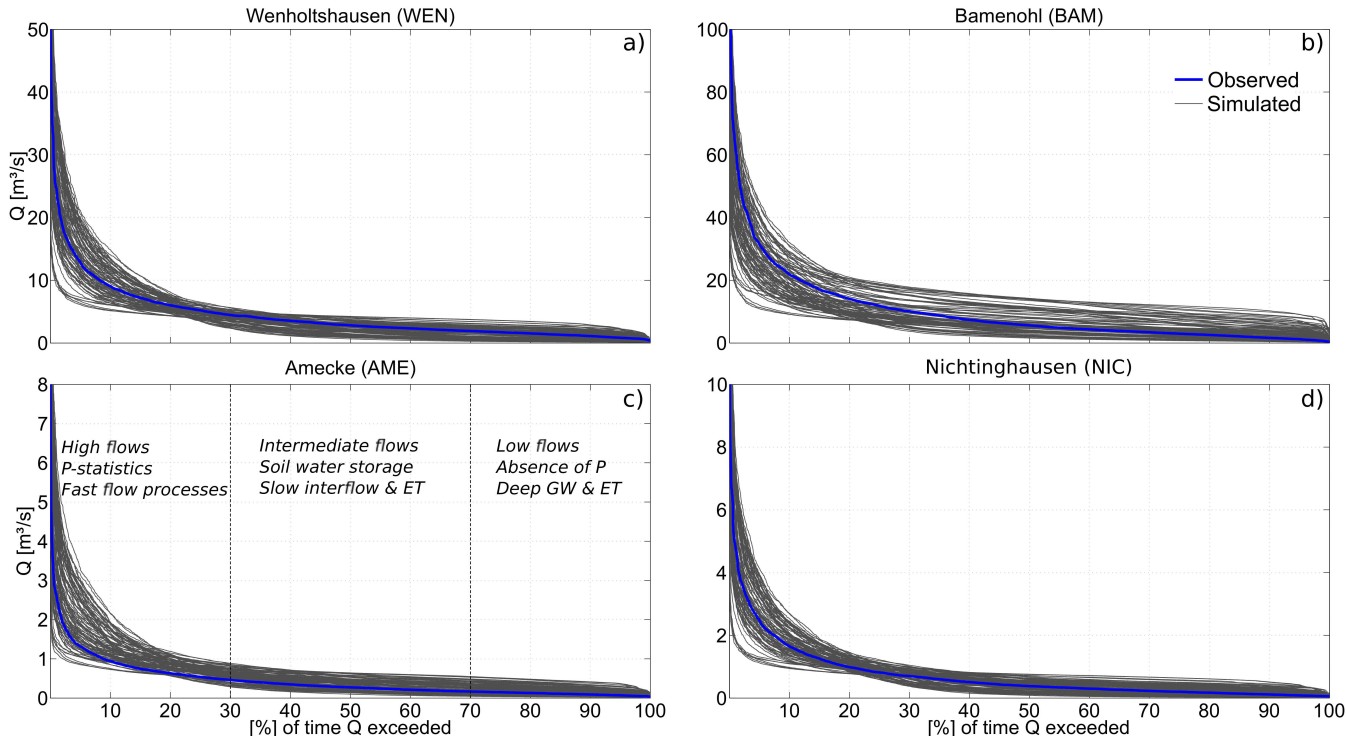

**Figure 3.** Ensemble of FAST-mHM simulated & observed FDCs of 4 selected Ruhr headwater catchments WEN (a), BAM (b), AME (c) and NIC (d) (daily flow, 2002-2011). Process controls in qualitative sequences of the FDCs proposed by Yilmaz et al. (2008) (c).





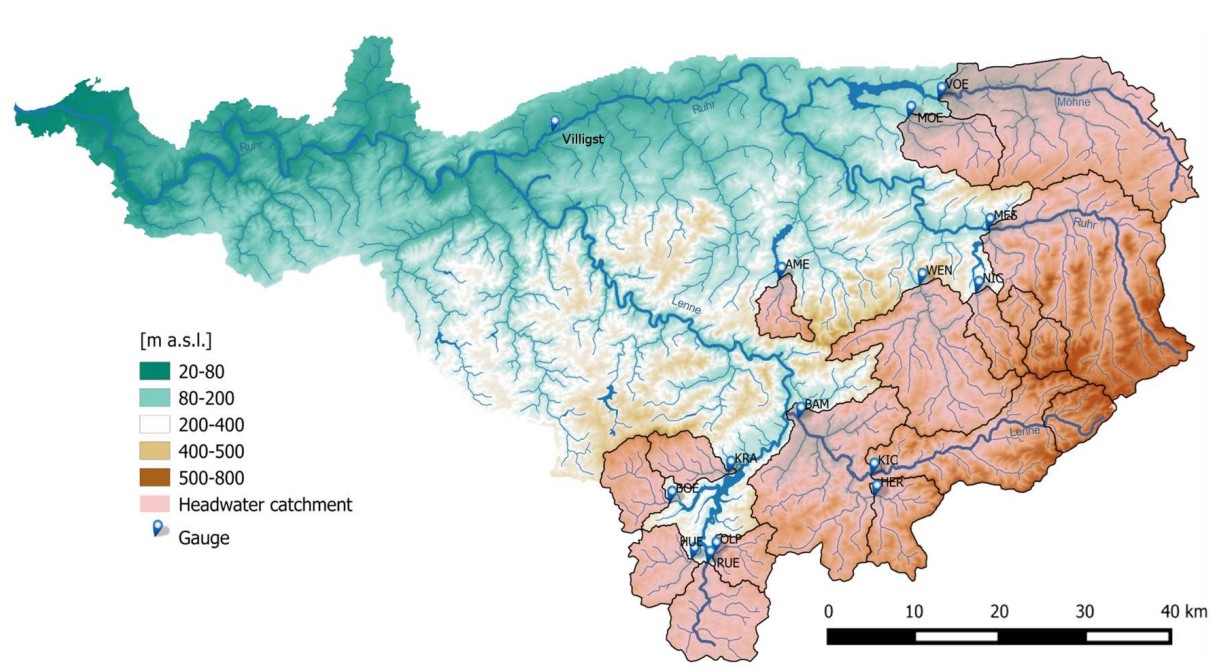

**Figure 4.** The Ruhr catchment with altitudinal zones, river network and 14 gauged headwater catchments and downstream gauge Villigst.





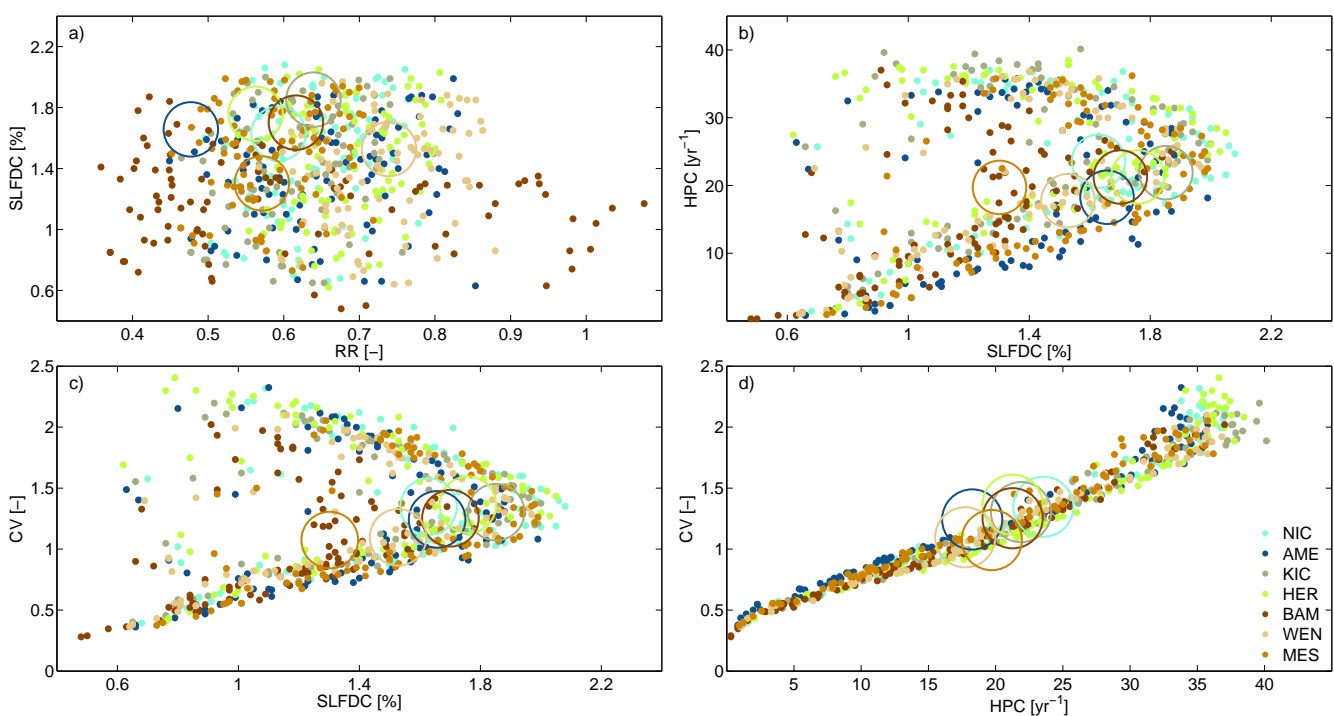

**Figure 5.** (a-d) Observed and scatterplots of the ensemble of FAST-mHM simulated dynamic response fingerprints. The centers of large circles depict observed values of combined response fingerprints. Circles' radii were scaled by the average values of the respective mean $\sigma$ of both fingerprints considered in each subplot qualitatively illustrating a possible constraint for the FAST-mHM parameter space.





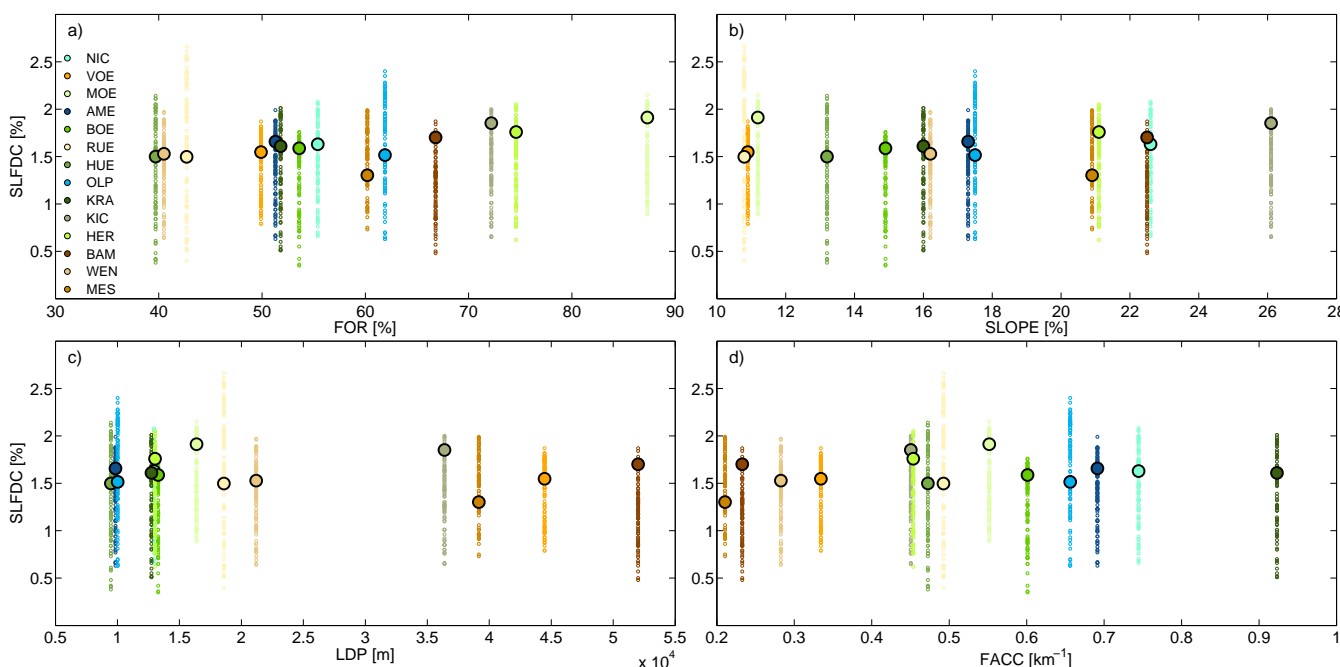

**Figure 6.** Scatterplots of observed (filled large circles) and ensemble of the simulated dynamic response fingerprint SLFDC against the physiographic fingerprints land cover type forest (FOR) (a), average catchment slope (SLOPE) (b), Longest Drainage Path (LDP) (c) & Flow Accumulation (FACC) (d) at 14 headwaters of the Ruhr basin. Circles of observed quantities indicate the sample space to be considered as constraint for the FAST-mHM parameter space.





**Figure 7.** Performance of most behavioural parameter sets for each dynamic response fingerprint at seven headwater gauges (a) and for Runoff Ratio (RR) and Slope of Flow Duration Curve (SLFDC) with corresponding ensembles of FAST-mHM simulated fingerprints at gauges AME and WEN (b). NSE against Euclidean distance ($D_E$) between four observed and ensemble simulated pairs of dynamic fingerprints (b-c). Spearman's rank correlation (e) and water balance error (f) against NSE for ensemble simulations with a focus on the overall 5 most behavioural parameter sets of 4 analysed pairs of fingerprints. Most behavioural parameter sets are plotted with their model run number and as large markers (a-f).







**Figure 8.** Comparison of streamflow hydrographs corresponding to different derivation: Most behavioural FAST-mHM parameter sets of 4 combinations of dynamic fingerprints, FAST-mHM parameter set with the highest Nash-Sutcliffe model efficiency (NSE) and observation at gauges WEN (a) and AME (b).





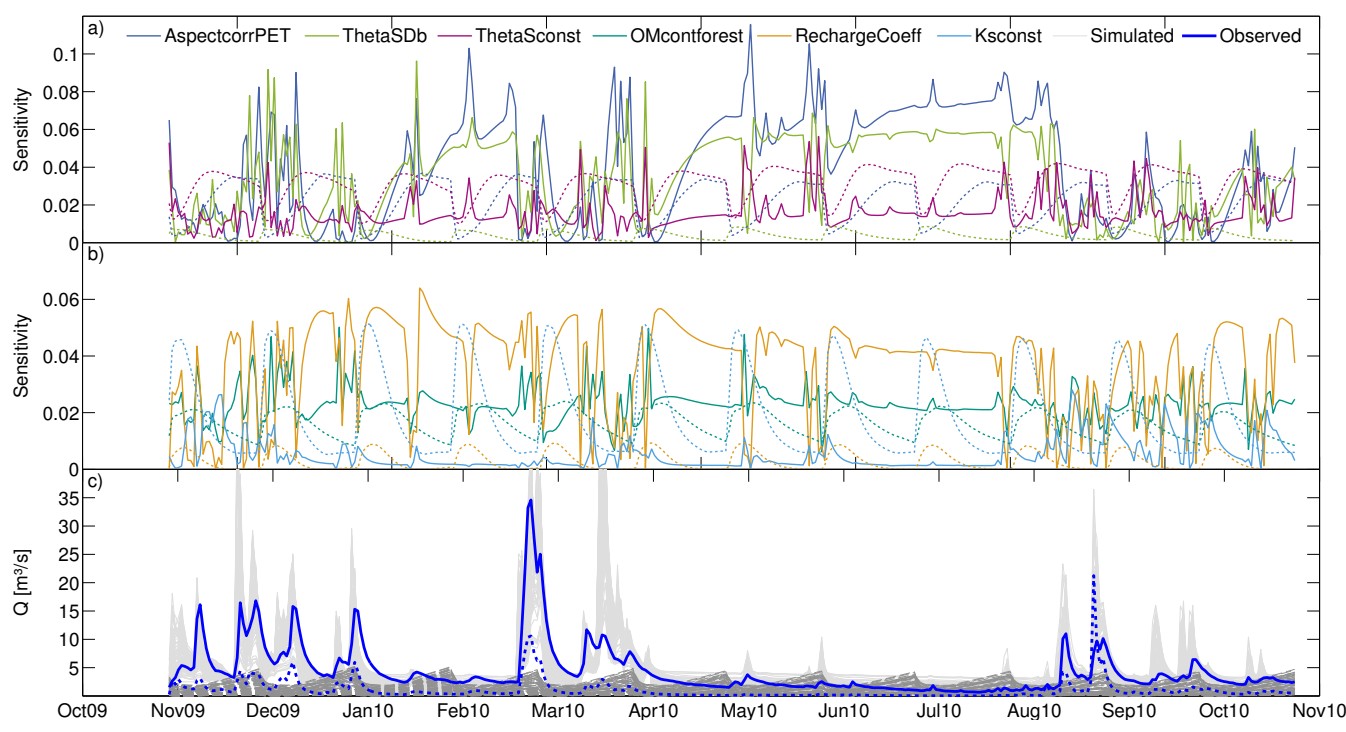

**Figure 9.** Time-dependent FAST sensitivities of six global mHM parameters for gauges WEN (solid lines) and HUE (dashed) (a and b). Observed and FAST-mHM simulated streamflow ensembles at gauges WEN (solid blue & light grey) and HUE (dashed blue & grey) (c).





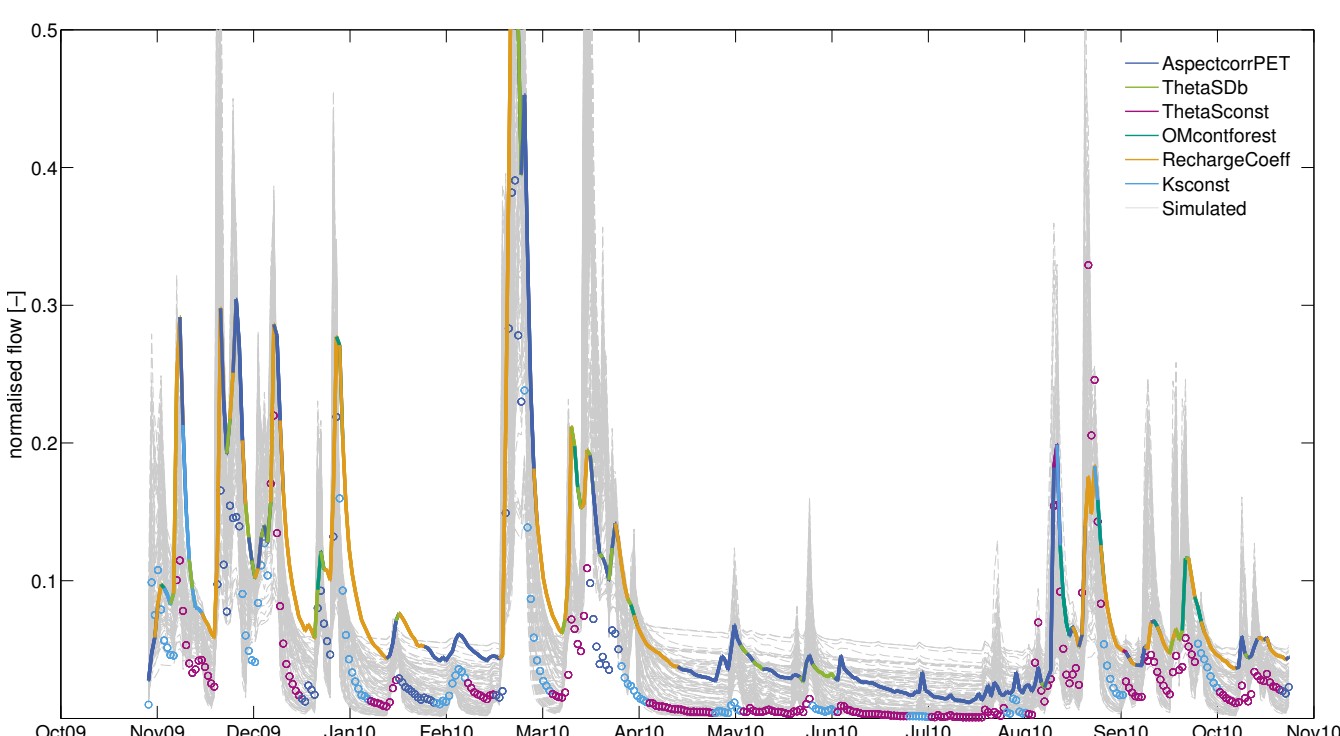

**Figure 10.** Highest parameter sensitivity related observed and corresponding ensemble of FAST-mHM simulated streamflow hydrographs at gauge Wenholthausen (WEN) (solid line) and highest parameter sensitivity related observed hydrograph at gauge Hüppcherhammer (HUE) (circles).





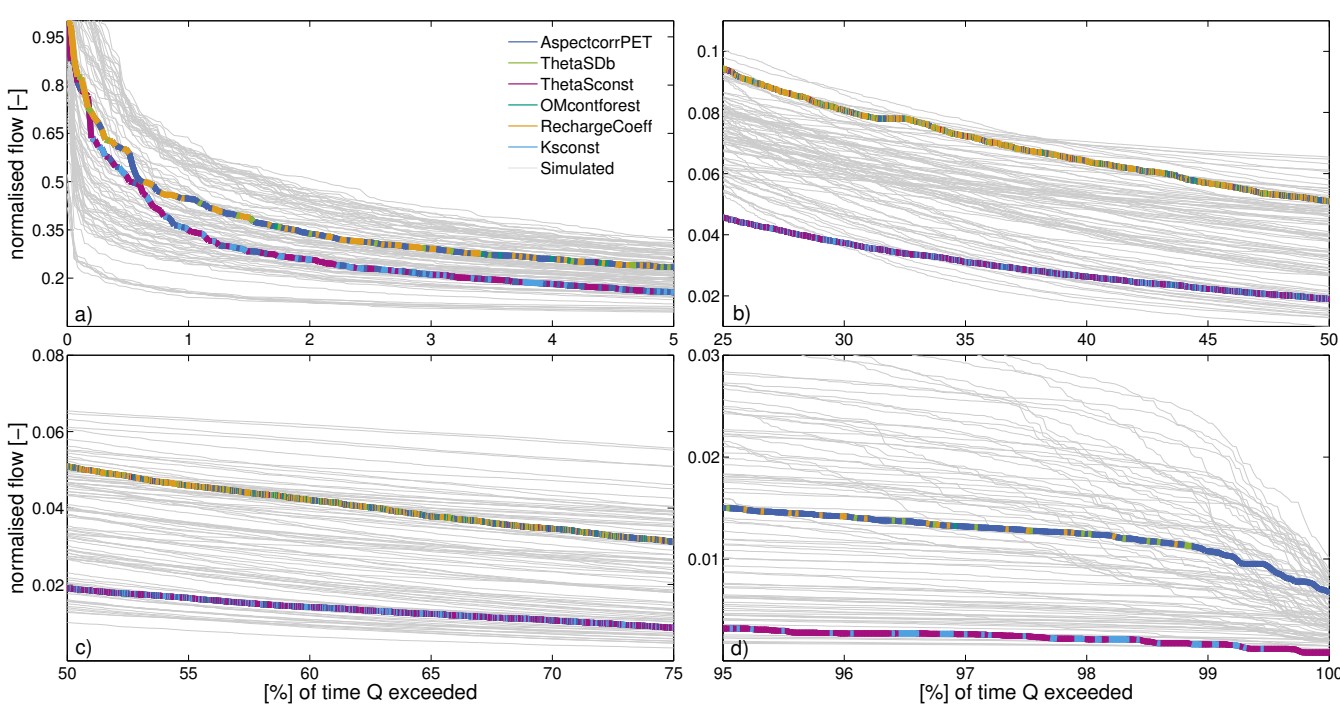

**Figure 11.** (a-d) 4 sequences of streamflow exceedance of highest parameter sensitivity related observed and corresponding ensemble of FAST-mHM simulated FDCs at gauge Wenholthausen (WEN) (upper FDC) and observed FDC at gauge Hüppcherhammer HUE (lower FDCs).





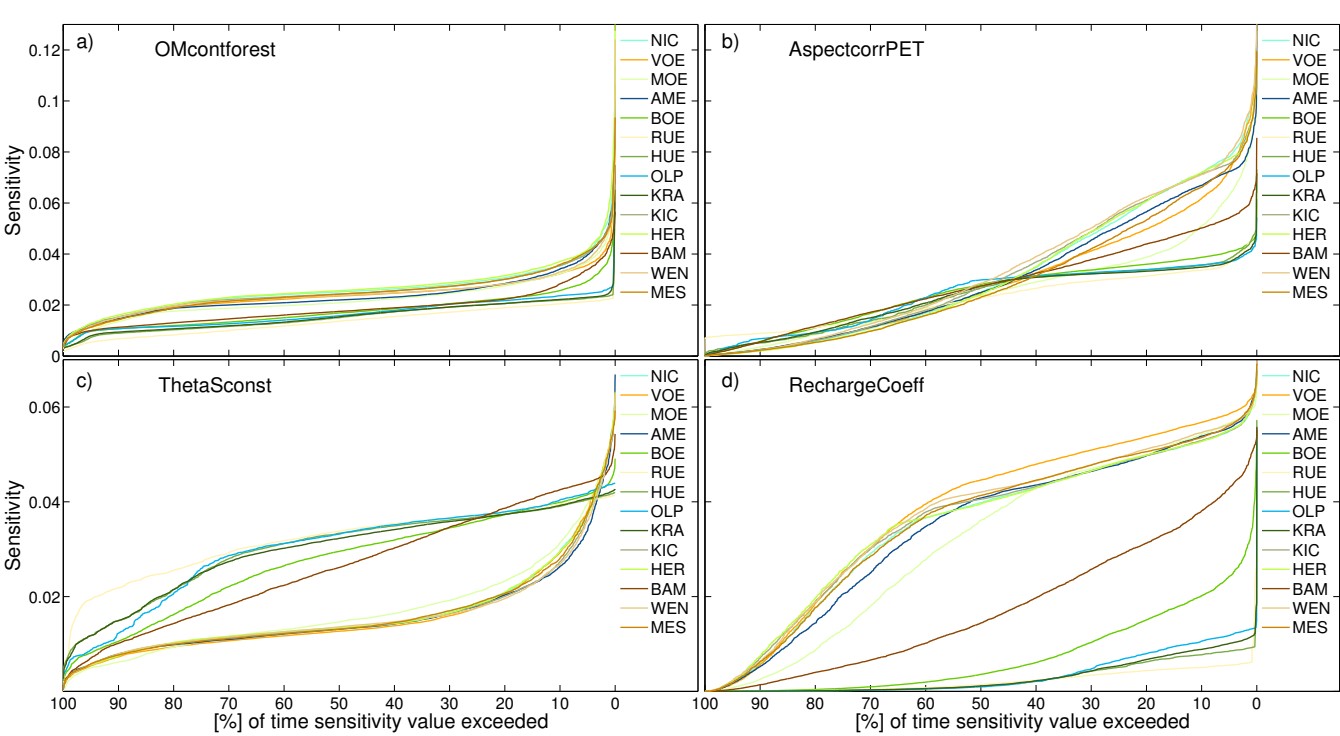

**Figure 12.** Sensitivity duration curves (SDCs) for 4 mHM parameters OMcontforest (a), AspectcorrPET (b), ThetaSconst (c) and Recharge-Coeff (d) in 14 Ruhr headwater catchments (2003-2011).

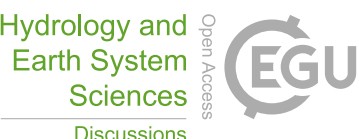

**Table 1.** Six global mHM parameters, function, value and FAST sensitivity ranges (2002-2011) for 14 Ruhr headwater catchments.

| Parameter | Process | Description | Value Range [-] | Sensitivity Range [-] |
|---|---|---|---|---|
| AspectcorrPET | meteo correction | Account for aspect $[°]$ dependent correction of PET $[mmd^{-1}]$ $[m^3m^{-3}]$ | 0.70 - 1.50 | 0 - 0.132 |
| ThetaSDb | soil moisture | Estimation of water content of saturated soil (Bulk density) $[gcm^{-3}])$ | -0.37 - -0.19 | 0 - 0.259 |
| ThetaSconst | soil moisture | Estimation of water content at saturation of soil (Constant part) | 0.65 - 0.95 | 0 - 0.067 |
| OMcontforest | soil moisture | Portion of organic matter [%] in soil to account for effects of organic matter | 0 - 20 | 0.002 - 0.176 |
| Ksconst | soil moisture | Estimation of saturated vertical hydraulic conductivity $[cmd^{-1}]$ | -1.9 - 0.0 | 0 - 0.074 |
| RechargeCoeff | percolation | Determination of percolation $[mmd^{-1}]$ coefficient $[d^{-1}]$ | 0 - 70 | 0 - 0.069 |



**Table 2.** Physiographic, climatic and dynamic response fingerprints to classify headwater catchments and constrain parameter space of mHM.

| Fingerprint Category | Functional Class | Hydologic Fingerprint |
|---|---|---|
| Physiography and climate | Climate | Wetness Index (P/ET) |
| | Landform | Area (A) |
| | | Longest Drainage Path (LDP) |
| | Topography | (Weighted) Slope (SLOPE, $I_g$) |
| | | Elevation Range |
| | | Flow Accumulation (FACC) |
| | Land cover | Forest (FOR), Urban (URB), |
| | | Pasture (PAST), Cropland (CROP) |
| | Soil | Baseflow Index (BFI) |
| Dynamic response | Average streamflow | Runoff Ratio (RR) |
| | | Total Q / Total P |
| | Streamflow variability | Coefficient of Variation (CV) |
| | | $\sigma/\mu$ |
| | Frequency of flow events | High Pulse Count (HPC) |
| | | Number of timesteps Q > 3*Q(mean) / years |
| | Rate of change in streamflow | Slope of Flow Duration Curve (SLFDC) |
| | | Between 33 % & 66 % exceedance |
| | High flow events | High Flow Discharge (HFD) |
| | | Q(5th percentile) / Q(median) |