# Peer review of "An integrated multi-fingerprint sensitivity-nested approach for regional model parameter estimation and catchment similarity assessment"

_Hydrology and Earth System Sciences, 2016_

## Referee Comment (RC1) · B. Guse (Referee) · 15 Jul 2016

Review of the manuscript "An integrated multi-fingerprint sensitivity-nested approach for regional model parameter estimation and catchment similarity assessment" by Höllering et al.

In this manuscript, Höllering et al. provide an approach to obtain a better understanding of model parameter behaviour and a more process-based model parameter estimation under consideration of spatial variations. For this, they used at first a fingerprint analysis to investigate how well the model performs for different aspects of the hydrological

system. Second, a temporally resolved parameter sensitivity is used to detect the dominant model parameters along the time series.

Overall, I really like the idea of this approach. However, I think that the core ideas of this study needs to be clarified. The introduction is not appropriate and needs to be reworked (including references). Furthermore, I think that the presentation of the results in the figures can be improved (also in their quality) and condensed, while the interpretation towards the overall benefit for the hydrological community can be enhanced.

Thus, I recommend a major revision of this manuscript. For this, I make several recommendations below.

MAJOR COMMENTS

Introduction:

The introduction needs to be completely reworked and restructured. I see here several reasons for this recommendation.

The introduction is not related to the abstract. It is very surprising that the introduction starts with catchment classification / similarity after reading the abstract. Further, in the methods and results, the performance (fingerprint) analysis and the temporal parameter sensitivity analysis have at least the same priority.

An introduction into the state-of-the-art in parameter sensitivity and more specifically on temporal parameter sensitivity analysis is completely missing. There are several studies in recent years using a temporally resolved sensitivity analysis and trying to extract helpful information for parameter understanding. Please see among others: Guse et al. (2014, 2016), Herman et al. (2013a, b), Massmann and Holzmann (2012), Massmann et al. (2014).

Furthermore, I am not satisfied with the introduction into performance analysis in terms of fingerprint analysis and parameter constraints. Also here, there are several recent

advances which need to be considered here to see this study in the context of the state-of-the art in research. Exemplarily please see: Euser et al. (2013, 2015), Gharari et al. (2014), Pfannerstill et al. (2014), Pokhrel et al. (2012), Reusser et al. (2009).

Thus, it is certainly required to extract in a better way the novelty of this approach in the context of temporal parameter sensitivity analysis, performance analysis and constraints for model parameters compared to the state-of-the art.

Objectives:

Following of my comments on the introduction, the objectives are not clearly enough motivated. Please check in the introduction whether all objectives are really motivated in the introduction. According to the current version of the introduction, I do not see a clear reason why it is relevant to constraint parameters in relation to different flow conditions or whether it is required to look at parameter sensitivity to understanding spatial distributed catchment behaviour. Certainly, I agree to both research questions, but I do not agree with their motivation.

Concept and methods:

According to my understanding of the manuscript, the three pillars are not really representing the article. The major point seems to be the third part. The second point is not very specific (streamflow generation). I would recommend to emphasize here more the general idea by shortly explaining how the different parts are related and which benefit is intended to obtain by the different steps. This is currently a bit unclear. Especially when presenting the concept, a clear structure is required. Maybe a flowchart would be more helpful than a list of five steps.

The link between the introduction and the concept is also not clear. Catchment classification dominates the introduction, but is not included in four of the five research steps.

The authors should think about directly considering the relationship between parameter

sensitivity and dynamic fingerprints. This aspect is somehow missing at the end and could provide helpful insights into parameter understanding.

Results:

There is a huge amount of figures showing the results. My impression is that due to the content of the figures (and the different subplots), the overall goal of this study is somehow lost. I strongly recommend to focus on figures showing the major outcomes.

In the results (5.3), it is not clear which knowledge is really gained by using the best selected model runs for a certain fingerprint (Fig. 7a). I think that we are more interested in overall best performing model runs combining different indices than in having a model runs which is the best in relation to a single fingerprint.

I had expected a presentation of a joined metric combining all fingerprints such as e.g. shown in Pfannerstill et al. (2014) or Haas et al. (2016), so that a model run could be finally selected which performs well for all fingerprints. In my opinion, this would be a reasonable final result of this part.

A discussion of the results in the context of the state-of-the-art and of how these results are related to knowledge obtained by former studies in this topic is missing. Concerning this, please see among others the list of references at the end.

I really like the expression Sensitivity duration curve (SDC). It is especially good to see that in this case differences between the catchments were detected. This is contrast to Guse et al. (2014) where a similar presentation (Fig. 6, even when it was not named Sensitivity duration curve) did not show relevant differences (due to a lack of spatial heterogeneity).

Maybe the authors could think about a final figure summarizing the results qualitatively. An example for this could be found in Fig. 9 in Herman et al. (2013a).

Discussion and conclusion:

Concerning the stated research questions, I think that the first research question is not really solved. I agree that fingerprints can help in constraining parameter ranges. However, this was also expected, since each hydrologic metric can somehow constraint parameter range. Here, I miss either a method to select overall behavioural parameter sets based on all (or all not correlated ones) fingerprints or a hydrological explanation that a certain fingerprint is able to constrain a parameter range since it represents the associated parameter accurately or something similar.

The sentence on P. 18, L. 26-27 "We further found..." really makes a strong difficulty apparent. The results that the parameter values (or constraints) are largely varying between the different behavioural parameter sets are problematic since it is then difficult to estimate the "best" parameter values. I encourage the author to discuss this point more in detail by suggesting possible way towards an overall behavioural parameter set.

In this context, I think that a more profound discussion of the relationship between the performance of fingerprints and model parameters would be helpful. At the end of the answer to research question 2 (P. 19, L. 6-16), the consistency between sensitivity duration curves and hydrologic fingerprints can be discussed. Do the spatial patterns of both are consistent?

Figures:

All figures: Overall, please check which are the most important figures and which are of less importance and could be removed/reduced.

In relation to this, several figures are not clear enough in terms of their intention and the visibility of the results. In particular, the description of the results in the figures (e.g. Figs. 5 and 7) is sometimes difficult to grasp. Concerning this remark, please see also the following comments to the figures.

Fig. 5: This figure needs to be improved. It is impossible to extract the information

of the relationship between same coloured points and the circle. One idea could be a reduction of the selected stations (in this plot) and/or a quadratic plot (increase of the plot height). Another idea could be to add a table (maybe as acknowledgement) stating how many points are within the circles. To summarize this comment: It should be possible to extract the information of how many points are in a circle somehow.

Fig. 6: I do not understand why and how the sample space is defined by the observed quantities in a fingerprint. Here, a clear approach is missing. I can agree that in the best case all simulations should be in the range as defined by the observed values of all stations. However, how is this related to the parameter space? This requires at least a detection of the parameter values leading to the fingerprint values as well as a clear relationship between parameter value and fingerprint. Maybe I understand something wrong here, but for me it seems to be that an information is missing here. It is really crucial for this study to understand this point.

Fig. 7a: This sub-figure needs to be completely reworked and improved. I cannot extract the relevant information. There is too much information: Seven sites as colours, six fingerprints as symbols and two metrics as well as the number of the best performing model run. One idea could be a plot in similar way as Figs. 1-6 in Bastidas et al. (2006). In this case, a separate plot could be shown for each performance metric.

MINOR COMMENTS:

Abstract, first sentence: I think that the first sentence should be more general. It is not clearly apparent why parameters need to be identified in relation to parameter sensitivity and catchment classification.

Abstract, second sentence: This sentence is certainly too long. Please subdivide this sentence into two (or even three) to avoid losing the reader directly at the beginning.

Abstract: The numbering (1), (2) and (3) is not explained and thus not understandable.

The referencing should be consistent. To give an example: On Page 5, Line 25, the

three references are neither ordered by occurrence nor alphabetic. Please check this in the whole text.

At several parts of the manuscript, the transitions between the different subchapters are not clear. I recommend to check the beginning and the end of the different chapter and if required add a sentence to relate both chapters. One example for this is the beginning of chapter 5.4.

P. 7 L. 3: Why do you use only six parameters which only explain less than the half of the variance? Later in the text (P. 15, L. 24), it is mentioned that even on the day with the highest sum of the partial sensitivities, this value is lower than 0.5. I think that a good reasoning for this is required. Even though that I am aware that it is not useful to do a temporal sensitivity analysis with all model parameters, I am curious whether it is possible to increase the explaining variance by using e.g. 8 or 10 parameters. Or otherwise, could you explain why a higher number of parameters is not beneficial/possible?

P. 7, L.11: I would recommend to write here: "In the case of using six parameters, the FAST method requires altogether 91 model runs..." It should be highlighted that the number of model runs depends on the model parameters and is provided by the FAST methods meaning that the same number of model runs is required for the same number of parameters independently from model, catchment or parameter selection.

P. 7, L.25: Here, 91 model runs are used to identify values for six parameters. This approach is certainly in contrast to typical model calibration algorithms using a significantly higher number of model runs for the same number of parameters. Even when I agree of using this approach for this study, I think that it is required to mention that a lower number of model runs is acceptable here according to capture all goals of this study. Or other way round, it is required to say that this number of model runs has to be certainly higher when only focusing on model optimization.

P. 8, L. 6: Which are the five classes?

P. 8, L. 17: There are more than six hydrologic fingerprints in Table 2. Could you explain why you mentioned here "six" hydrologic fingerprints?

P. 8, L.19: Which one (of the dynamic fingerprint)?

P. 8, L. 26: I would recommend to write: "from each of the simulated..."

P. 8, L. 30: I think that here and maybe also in the introduction a discussion of the PAWN method is missing as proposed by Pianosi and Wagener (2015) since the role of different performing model results within the sensitivity analysis is directly included in PAWN.

P. 9, L.-10-25: I recommend to also refer here to the work from Pfannerstill et al. (2015) and Pokhrel et al. (2012). In both studies FDC and their segments are used to identify (constrain) parameter values.

P. 11, L. 12-13 (Fig. 5): Could you explain why you used circles assuming that the variation are similar for both variables. Is it maybe more useful to use an ellipse (which it would be in the case of a quadratic plot)?

P. 11, L. 12-16: I did not understand how the radius is selected and why it has this size.

P. 11, L. 21: Could you explain why you mentioned both distance measures? Is it maybe more appropriate to select one (the best) of them?

P. 12, L. 21-28: Which result (figure) supports this text passage?

P. 13, L. 20: Please explain why you have selected these seven gauging stations

P. 14, L. 2: Why do you selected four fingerprints and calculate an Euclidean distance between them. Why not using all fingerprints?

P. 14: I have expected a clearer description of the intention of each subplot of Fig. 7 and a presentation of the major outcome of each subplot.

P.16, L. 26: Please add here or later in the discussion that the relationship of parameter

sensitivities and FDC (or sorted discharge) was already captured e.g. in Herman et al. (2013b) and Guse et al. (2016) if not already included in the introduction after revision.

P. 16, L. 29: Is there a reason why two observed and only one simulated station are used in Fig. 11?

P. 20, L. 21: I would recommend to structure the discussion in two sub-chapters to avoid a misunderstanding evoked by a double-use of the numbers 1-3 in the discussion. The second sub-chapter in the discussion could start at this line.

P. 21, L. 12: Why not directly increasing the number of parameters in this study?

Fig. 1: I do not see the relationship between the FAST sampling design the parameter values in the calibration. Why do you show both in one plot? Which information can be derive from this relationship? Furthemore, due to the different ranges of the parameters, the interpretation of the parameter values is rather difficult.

Fig. 1, caption: Please changed to "parameter values in the 91 model runs according to the FAST sampling" or a similar expression.

Fig. 2: Please increase the labels a and b in the figure.

Fig. 2: I strongly recommend to subdivide this figure into two plots showing separately hydrologic fingerprints and the dynamics response.

Fig. 2b: The lines in the dynamic response fingerprints are unclear.

Fig. 3: Do you really need this figure? I do not see the real benefit.

Fig. 3: Yilmaz et al. (2008) made a FDC segmentation at 20% and not at 30% of flow exceedance.

Fig. 3: Are you showing here the 91 model runs as FDC? In this case it would suggest to clarify this by stating this.

Fig. 4: Since the gauges and their abbreviation are used several times in the

[Figure]

manuscript, I strongly recommend to increase the labels in size. Maybe a white background (for the labels) would be helpful in addition.

Fig. 7: Please think about the benefit of each subplot.

Fig. 7: The legend to the gauges belongs to Fig. 7a and not 7f.

Fig. 7b: Please discuss in the text why the best performing run in relation to SLFDC is among the worst runs related to NSE.

Fig. 8: Please add in the figure caption that the numbers in brackets in the legend are the numbers of the model runs.

Fig. 10: Please explain in a better way: "highest parameter sensitivity related observed hydrograph".

Fig. 11: It seems to be that the major information from these plots could be extracted in a simpler way. I do not think that the grey lines are required. What about showing only the changes in the dominant parameters as a line (or a row) for each gauge.

Fig. 12: Maybe the legend could be shown only once and outside of the plot at the right side (only a very minor comment).

Technical corrections: P.1, L. 10: sensitivity P.4, L. 16-18: This sentence does not read well. P. 5, L.6: It recommend to use the paper of Reusser et al. (2011) instead of the dissertation work (Reusser 2010).

References:

Bastidas LA, Hogue TS, Sorooshian S, Gupta HV and Shuttleworth WJ (2006): Parameter sensitivity analysis for different complexity land surface models using multicriteria methods, J. Geophys. Res., Vol.11, D20101, doi:10.1029/2005JD006377.

Euser T, Winsemius HC, Hrachowitz M, Fenicia F, Uhlenbrook S and Savenije HHG (2013): A framework to assess the realism of model structures using hydrological signatures, Hydrol. Earth Syst. Sci, 17(5), 1893-1912.

Euser T, Hrachowitz M., Winsemius HC and Savenije HHG (2015): The effect of forcing and landscape distribution on performance and consistency of model structures, Hydrol. Process. 29(17), 3727-3743.

Gharari S, Shafiei M, Hrachowitz M, Kumar R, Fenicia F, Gupta HV and Savenije HHG (2014): A constraint-based search algorithm for parameter identification of environmental models, Hydrol. Earth Syst. Sci., 18, 4861-4870.

Guse B, Reusser DE and Fohrer N (2014): How to improve the representation of hydrological processes in SWAT for a lowland catchment – temporal analysis of parameter sensitivity and model performance. Hydrol. Process. 28: 2651–2670.

Guse B, Pfannerstill M, Strauch M, Reusser D, Lüdtke S, Volk M, Gupta H and Fohrer N (2016): On characterizing the temporal dominance patterns of model parameters and processes, Hydrol. Process., 30(13), 2255-2270.

Haas M, Guse B, Pfannerstill M and Fohrer N (2016): A joined multi-metric calibration of river discharge and nitrate loads with different performance measures, J. Hydrol., 536, 534-545.

Herman JD, Kollat JB, Reed PM and Wagener T (2013a): From maps to movies: high resolution time-varying sensitivity analysis for spatially distributed watershed models. Hydrol. Earth Syst. Sci. 17: 5109–5125.

Herman JD, Reed PM and Wagener T (2013b): Time-varying sensitivity analysis clarifies the effects of watershed model formulation on model behavior. Water Resour. Res. 49. DOI:10.1002/wrcr.20124.

Massmann C and Holzmann H (2012): Analysis of the behavior of a rainfall–runoff model using three global sensitivity analysis methods evaluated at different temporal scales. J. Hydrol. 475: 97–110.

[Figure]

Massmann C, Wagener T and Holzmann H (2014): A new approach to visualizing time-varying sensitivity indices for environmental model diagnostics across evaluation timescales. Environ. Model. Softw. 51: 190–194.

Pfannerstill M, Guse B and Fohrer N (2014a): Smart low flow signature metrics for an improved overall performance evaluation of hydrological models. J. Hydrol. 510: 447–458.

Pianosi F and Wagener T (2015): A simple and efficient method for global sensitivity analysis based on cumulative distribution functions, Environ Model Softw. 67, 1-11.

Pokhrel P, Yilmaz KK and Gupta HV (2012): Multiple-criteria calibration of a distributed watershed model using spatial regularization and response signatures. J. Hydrol. 418: 49–60.

Reusser DE, Blume T, Schaefli B and Zehe E (2009): Analysing the temporal dynamics of model performance for hydrological models. Hydrol. Earth Syst.Sci. 13: 999–1018.

---

## Referee Comment (RC2) · S. Gharari (Referee) · 18 Jul 2016

After reading the manuscript, I must admit that "I am lost and confused"! It seems the manuscript is revolving around many but at the same time no clear message. Either I missed or I am not capable to fully understand; I don't see any conclusion from the presented manuscript!

Starting from the structure; the manuscript is very badly structured and too wordy and long. The literature review is spread over the paper and there are a lot of unnecessary sub-sections and subsub-sections which can be reduced.

[Figure]

The title looks very sophisticated and broad. Every word need further explanation which the manuscript fails to fully explain. As an example, "integrated" and "sensitivity-nested" seem to have similar meaning. "Multi-fingerprint" can be replaced by much simpler words. Did the authors really carry out a "catchment similarity assessment" as stated in the title?

The abstract is again distance from what the paper is trying to tell. One cannot really understand the final conclusion of the paper from the abstract and only an unclear and again broad sentence such as "The sensitivity approach may be useful..." concludes the abstract. I would also remove the world "novel" from the abstract and any other places in the manuscript. It is the readers' decision to decide about the novelty not the authors'.

The introduction is very wide too with many relevant and irrelevant studies put together next to each other in non-coherent sentences. The general literature review is not enough and missing a significant body of the existing literature on catchment similarities, catchment classification and so on. As an example, one of my own research interests which have been mentioned in section 1.2 can be re-written for topographical landscape unit as follow:

"Using indices based on topography can be one of many ways to delineate a catchment into hydrological response units or as stated by Zehe et al., 2014, functional units. These units can be built based on topographical features as stated by Knudsen et al., 1987, Flugel 1995 and winter 2001 into X1, X2 and X3. Aligned with what have been suggested and with help of a recent topographical index (HAND, Renno et al., 2008) Gharari et al (2011) classified a small (or meso scale) catchment in Luxembourg and then used the mapping for building and constraining a conceptual model (Gharari et al 2014a). This classification have been repeated by Gao et al 2014 for a large scale catchment in China and served as the basis of the modeling exercise."

The authors can extend this section by elaborating on slope, soil and land cover.

Please keep in mind that the introduction, as well as each paragraph, should have a funnel shape structure starting from broad and general overview and ending in example of specific implications. This ways the reader will be ready for the final and general message of the paper.

"Is the inconsistency of functional unit and physiographic similarities a paradox?" No one really claim that response units should exactly follow physiography. This is just an assumption to give us the ability to make of prediction of what we really don't know (such as prediction in ungauged basin).

I encourage the authors to give a comprehensive literature review in section 1.4 regarding the signatures (or fingerprints). I would also ask the authors to give a comprehensive literature review on the use of signature in the hydrological modeling (such as Euser et al., 2013 and Clark e t al., 2011). What remains missing in the introduction given the gist of title and current introduction is the relation between the model, signature and processes. There have been numerous studies looking on this topic, the sensitivity and also uncertainty of the model parameters over different period of time. I remember reviewing a manuscript on this topic for the very same journal HESS, Pfannerstill et al., 2015. Dr. Guse gave a very wide range of publication related to this in his review.

Section 3.4.2 is a mixture of literature review of the signatures, and FAST and selected signature for this study. I would advise the authors to briefly clarify what signatures they used in this study. The literature review should have been presented earlier and FAST is explained later in the manuscript which makes it a bit difficult to follow. I would also suggest to clarify the possible limitation of FAST compare to the other sensitivity analysis, Razavi and Gupta (2015) may help in that regard.

Please clarify the name and possibly the similarities of the headwater catchments in section 4.1 and 4.2. The names of the headwaters pop up every now and then in the manuscript!

[Figure]

About the parameter constraining: In my point of view what the authors are presenting in this study is not parameter constraining, it is rather parameter selection given various signature and based on sensitivity analysis. We had recent papers looking at constraining (Gharari et al., 2014a and 2014b, Hrachowitz et al., 2014). There are many related existing studies as well, the authors are more than welcome to address in them in their manuscript. Constraining should be different in my point of view from parameter selection based on a multi objective approach (which is similar to breaking the evaluation of the time series into different signatures).

The figures and result part is very hard to follow. I believe one of the main reason is the simultaneous presentation of the results together with the methods. I would have separated the method and result sections. The figures are almost very difficult to follow, as an example figure 5, 6, 7, 8, and 11. Again back to what I mentioned earlier the authors are not constraining the parameter sets but they are selecting different parameter sets based on different criteria (Figure 7). The criteria the authors are using to me seems like a multi-objective approach where the most balanced parameter sets are selected by Euclidian distance. We did this in our study in 2013 as well (Gharari et al, 2013) using Pareto front members and Euclidian distance to pick the behavioral parameter sets. The point of that study was not to constrain the model parameters but to show that the parameter sets which are better performing over time are different from the optimal or behavioral parameter sets.

Conclusion and discussion is vague as well. I would suggest the authors to separate the conclusion and discussion parts. If the paper have a message and strong conclusion the conclusion part should be only few bullet points, pointing at the most general and specific findings of the manuscript. The authors can make use of this bullet point conclusions to form the abstract, introduction and methodology better.

The manuscript is far from the minimum quality for publication, meaning that it should be rejected. To give a chance for the manuscript contribution to be published in HESS I would give major revision. I am looking forward to receiving the revised manuscript. I

hope my comments help the authors to elevate the level of the current manuscript and present in much better shape.

With kind regards

Shervan Gharari

PS. I thank Fuad Yassin who helped me better understand the manuscript. Our discussion was helpful for writing this review. I guess he was also "lost and confused"! ;)

References:

Gharari, S., Hrachowitz, M., Fenicia, F., and Savenije, H. H. G.: An approach to identify time consistent model parameters: sub-period calibration, Hydrol. Earth Syst. Sci., 17, 149–161, doi:10.5194/hess-17-149-2013, 2013.

Gharari, S., Hrachowitz, M., Fenicia, F., Gao, H., and Savenije, H. H. G.: Using expert knowledge to increase realism in environmental system models can dramatically reduce the need for calibration, Hydrol. Earth Syst. Sci., 18, 4839-4859, doi:10.5194/hess-18-4839-2014, 2014.

Gharari, S., Shafiei, M., Hrachowitz, M., Kumar, R., Fenicia, F., Gupta, H. V., and Savenije, H. H. G.: A constraint-based search algorithm for parameter identification of environmental models, Hydrol. Earth Syst. Sci., 18, 4861-4870, doi:10.5194/hess-18-4861-2014, 2014.

Gharari, S., Hrachowitz, M., Fenicia, F., and Savenije, H. H. G.: Hydrological landscape classification: investigating the performance of HAND based landscape classifications in a central European meso-scale catchment, Hydrol. Earth Syst. Sci., 15, 3275-3291, doi:10.5194/hess-15-3275-2011, 2011.

Euser, T., Winsemius, H. C., Hrachowitz, M., Fenicia, F., Uhlenbrook, S., and Savenije, H. H. G.: A framework to assess the realism of model structures using hydrological

signatures, Hydrol. Earth Syst. Sci., 17, 1893-1912, doi:10.5194/hess-17-1893-2013, 2013.

Hrachowitz, M., Fovet, O., Ruiz, L., Euser, T., Gharari, S., Nijzink, R., Freer, J., Savenije, H. H. G., and Gascuel-Odoux, C.: Process consistency in models: The importance of system signatures, expert knowledge, and process complexity, Water Resour. Res., 50, 7445–7469, doi:10.1002/2014WR015484, 2014

Gao, H., Hrachowitz, M., Fenicia, F., Gharari, S., and Savenije, H. H. G.: Testing the realism of a topography-driven model (FLEXTopo) in the nested catchments of the Upper Heihe, China, Hydrol. Earth Syst. Sci., 18, 1895–1915, doi:10.5194/hess-18-1895- 2014, 2014.

Zehe, E., Ehret, U., Pfister, L., Blume, T., Schröder, B., Westhoff, M., Jackisch, C., Schymanski, S. J., Weiler, M., Schulz, K., Allroggen, N., Tronicke, J., van Schaik, L., Dietrich, P., Scherer, U., Eccard, J., Wulfmeyer, V., and Kleidon, A.: HESS Opinions: From response units to functional units: a thermodynamic reinterpretation of the HRU concept to link spatial organization and functioning of intermediate scale catchments, Hydrol. Earth Syst. Sci., 18, 4635-4655, doi:10.5194/hess-18-4635-2014, 2014.

Flügel, W.-A.: Delineating hydrological response units by geographical information system analyses for regional hydrological modelling using PRMS/MMS in the drainage basin of the River Bröl, Germany, Hydrol. Process., 9, 423–436, doi:10.1002/hyp.3360090313, 1995.

Rennó, C. D., Nobre, A. D., Cuartas, L. A., Soares, J. V., Hodnett, M. G., Tomasella, J., and Waterloo, M. J.: HAND, a new terrain descriptor using SRTM-DEM: Mapping terrafirme rainforest environments in Amazonia, Remote Sens. Environ., 112, 3469–3481, doi:10.1016/j.rse.2008.03.018, 2008.

Martyn P Clark, Hilary K McMillan, Daniel BG Collins, Dmitri Kavetski, Ross A Woods, Hydrological field data from a modeller's perspective: Part 2: process‐based evaluation of model hypotheses, Hydrological Processes, 25, 4, 523-543, 2011

Fenicia, F., D. Kavetski, H. H. G. Savenije, and L. Pfister, From spatially variable streamflow to distributed hydrological models: Analysis of key modeling decisions, Water Resources Research(52), 1-36, doi:10.1002/2015WR017398, 2016.

Pfannerstill, M., Guse, B., Reusser, D., and Fohrer, N.: Process verification of a hydrological model using a temporal parameter sensitivity analysis, Hydrol. Earth Syst. Sci., 19, 4365-4376, doi:10.5194/hess-19-4365-2015, 2015.

Razavi, Saman, and Hoshin V. Gupta. "What do we mean by sensitivity analysis? The need for comprehensive characterization of "global" sensitivity in Earth and Environmental systems models." Water Resources Research 51.5, 3070-3092, 2015.

Knudsen, J., Thomsen, A., and Refsgaard, J. C.: WATBAL A SemiDistributed, Physically Based Hydrological Modelling System, Nord. Hydrol., 17, 347–362, 1986.

Winter, T. C.: The Concept OF Hydrologic Landscapes, J. Am. Water Resour. Assoc., 37, 335–349, doi:10.1111/j.1752-1688.2001.tb00973.x, 2001.

---

## Referee Comment (RC3) · F. Sarrazin (Referee) · 18 Jul 2016

In this manuscript, the authors present a parameter estimation and sensitivity analysis scheme to learn about differences in catchment functioning and to classify catchments. However, many points need clarification, in particular the implications and conclusions of the work. I recommend major revisions of the manuscript.

Parts of the manuscript appear to be quite long (specifically the introduction and results sections) and the reader tends to get lost. A general recommendation to the authors is to select in each section the key elements that contribute to their argumentation and

to clearly connect them. The authors should also try to keep their sentences short to improve readability.

Furthermore, critical information on the implementation of sensitivity analysis is missing (output considered, how the sample size was chosen), which makes it difficult to interpret the sensitivity analysis results.

I provide below more detailed comments for the different sections and some minor comments at the end.

SECTION 1 introduction

1) I think the introduction section is unnecessarily long and not sufficiently linked to the objective section (section 2). Specifically, how do the work of Bardóssy (2006) (p2 L11) Wagener et al. (2007) (p2 L18), Castiglioni et al. (2010) (p4 L14), Yadav et al. (2007) (p4 L18) relate to the work presented in the manuscript? In which way the literature review on catchment classification (section 1.2) is useful to better understand the work presented in the manuscript? I suggest the authors select key elements in the literature review and clearly link them to their study, so to clarify what the contributions of the manuscript are.

2) There is some confusion in Section 1.3 and it needs clarification. A distinction has to be made here between a model and the underlying system it represents. The term equifinality commonly refers to the model and not the system it represents. In fact, Beven (2006) (p21, paper cited by the authors) characterizes the term equifinality as the 'rejection of the assumption that a single correct representation of the system can be found given the normal limitations of characterisation data.' Therefore, equifinality is not an 'inherent' property of a model equations as stated by the author p3 L31, but it is due to the fact that the information content in the data is not sufficient to identify a unique model representation. I am also asking the question: Is the study of physical and hydrological similarities based on observation data or on model simulation outcomes?

3) p4, L20: the expression 'to raise the information considered in the parameter transfer' is quite fuzzy. Please clarify.

SECTION 2 Objective and driving research questions

1) As mentioned above, the objectives need to be clearly linked to the literature review, to show the contributions of the manuscript.

2) p4 L30-31: This objective need clarification. First, 'consistent manner' is vague and it is required to better explain this expression. Second, the motivation need to be clarified. Is the objective to learn about the model ability to reproduce the data, to learn about the value of observation data etc.? 3) p5 L8-9: the expression 'within the range of their independent variable' needs clarification.

SECTION 3 Concept and methodological approach

1) I have reservations regarding the implementation of the sensitivity analysis (section 3.3), specifically:

- p7 L12: How was the sample size chosen? It is necessary to check the convergence of the results, that is to say to assess to what extent the results would change if using a new sample. When the sample size is too small, the sensitivity analysis results can be unreliable (e.g. Sarrazin et al., 2016). A way to assess convergence is to derive confidence intervals on the sensitivity indices using bootstrapping. However, this technique cannot be easily applied when using FAST, given the structure of the sample. Therefore, I would suggest to simply look at the stability of the results when using smaller/larger sample sizes.

- It is required to specify the scalar output used for sensitivity analysis, otherwise the results are meaningless. Sensitivity analysis results can strongly vary when considering different model outputs (e.g. van Werkhoven et al., 2008).

- Why analysing main effects (FAST) only and not total effects (eFAST)? A parameter could have an effect through interactions only and therefore would not be detected

when applying FAST.

2) In Section 3.4.1 several points need clarification:

- p8 L3: 'sensitivity confined' is unclear.

- p8 L4: this sentence is vague

- p8 L5: this sentence have to be revised in relation to section 1.3.

3) p9 L10-25: This is quite a long paragraph and I am not sure in which way it connects to the work presented in the manuscript. Again, I suggest to select the relevant information that helps to better understand the present study.

SECTION 4 Study area

1) The authors should present the data used for the simulations and precise the simulation time horizon chosen for the analyses. Also, why presenting some of the data used before presenting the study area? (section 3.3.2, p7 L15-21)

SECTION 5 Results

1) A general comments for this section is that some figures are redundant and could be removed for the sake of brevity (e.g. Figure 6, 9 and 10). Likewise, the text could be more concise.

2) p11 L13: Why is it a 'qualitative constraint'?

3) p11 L29-30: The behavioural subset is the lower branch of the 'arrow' or the points that are within the circles?

4) p12 L2-3: What are the implications of the statements 'which portends [. . .] specifically.' and p12 L10-11 'Circles [. . .] varying strength'?

5) How does section 5.3 relates to section 5.1 which is also about constraint on parameter space?

6) In section 5.3, I think it may be more appropriate to identify for each fingerprint and pair of fingerprints not a unique most behavioural parameter set but an ensemble of best performing parameter sets. In fact, observation data are affected by uncertainties and it may not be relevant to make distinctions among the top performing parameter sets.

7) Why keeping correlated fingerprints in the analysis of section 5.3? For instance, CV or HPC could be remove from the analysis since the two fingerprints have the same information content.

8) p14 L12: Please clarify how the five most behavioural parameter sets were determined.

9) p15 L24-26: This statement is not correct. When referring to Sobol' variance decomposition (Sobol', 1990), the sum of main effects is equal to 1 when main effects only contribute to the total variance (no interactions). A sum of main effects for the six parameters analysed equal to 0.43 means that a significant fraction of the total variance (1-0.43=0.57) is due to interactions between parameters. If more parameters were added to the analysis, this would not necessarily result in an increase in the sum of the main effects, since the output variance woul also change. The same applies for the statement p21 L11-12.

10) Section 5.4.1: I have reservations regarding the interpretation of the sensitivity analysis results:

- p16 L4-10: I do not see any clear summer/winter pattern for the sensitivity of Aspect-corrPET and Recharge Coeff for WEN but more 'ups and downs' all year round.

- p16 L26-35 - p17 L1-11: I am not sure about the significance of Figure 11. The authors consider the most influential parameter only. However, the difference in sensitivity among the most sensitive parameters may be small and even not statistically significant given the approximation error in the sensitivity index values. This is all the

more concerning, since all the sensitivity indices take quite small values from Figure 9 (below 0.12).

11) p17 L10-11: The sentence 'Nevertheless [. . .] on streamflow generation.' is very fuzzy. It is required to clarify. I am also still wondering why the authors chose to study the main effects only and not the total effects.

SECTION 6 Discussions and Conclusions

1) I suggest splitting this section in two, with a discussion section and a concise conclusion section. This would help to highlight the contributions and implications of the work.

2) p18 L33: Either an indicator is normalised or it is not normalised but it cannot be 'less normalised'.

3) p19 L10-14 What are the implications of the statements 'The composition [. . .] in the eastern headwaters.'?

4) p19 L18-19 'by model [. . .] of meteorological forcing data' is quite fuzzy. Please clarify.

5) p20 L22: The authors do not actually present any result on catchment classification

MINOR COMMENTS

- title of section 1.3: replace 'inconconsistency' by 'inconsistency'

- there is an error in the reference to Beven (1990). Are the authors referring to

Beven, K: Changing ideas in hydrology - The case of physically based models, Journal of Hydrology, 105, 157-172, doi: 10.1016/0022-1694(89)90101-7, 1989.

Or

Loague, K.: Changing ideas in hydrology - The case of physically based models - Comment, Journal of Hydrology, 120, 405–407, doi:10.1016/0022-1694(90)90161-P,

http://linkinghub.elsevier.com/retrieve/pii/002216949090161P, 1990.

- p5 L8: replace 'of the selected fingerprints' by 'to the selected fingerprints'

-p5 L12 and L23: what do the authors mean by 'dependent'?

- p5 L13: replace 'of simulated streamflow and of the related fingerprints' by 'to simulated streamflow and to the related fingerprints'.

- p6 L21-23: Why introducing eFAST here if it is not used? I think the sentence can be removed to keep the manuscript concise, unless eFAST is actually used.

-p8 L19: aren't there five dynamic fingerprints?

-p10 L14-15: it is required to reformulate this sentence.

- p14 L31: replace 'to small' by 'too small'

- p16 L25: I don't think the term 'interacting inversely' is correct since only the main effects (and not parameter interactions) are analysed here. Please reformulate.

REFERENCES

Sarrazin, F., F. Pianosi, and T. Wagener. 2016. "Global Sensitivity Analysis of Environmental Models: Convergence and Validation." Environmental Modelling & Software 79: 135–52. doi:10.1016/j.envsoft.2016.02.005.

Sobol', I.M. 1990. "Sensitivity Estimates for Nonlinear Mathematical Models." Matematicheskoe Modelirovanie 2, 112-118 (in Russian), Translated in English (1993). In: Mathematical Modelling and Computational Experiments 1: 407–14.

Van Werkhoven, K., T. Wagener, P. Reed, and Y. Tang. 2008. "Characterization of Watershed Model Behavior across a Hydroclimatic Gradient." Water Resources Research 44 (1): 1–16. doi:10.1029/2007WR006271.

---

## Referee Comment (RC4) · Anonymous Referee #4 · 19 Jul 2016

I have to admit that I struggled for a couple of days to understand the message of this paper, and I am disappointed to say I failed to do so. My understanding is this paper is an amalgam of sensitivity analysis, parameter estimation and catchment clustering, however it is not well described how these approaches are linked together. Each of these elements, if performed elaborately, can be a separate paper and mixing them only confuses readers. Neither the abstract nor the introduction sections support the goals of the study. Well to be fair, goals are not clear either! Also I should mention that manuscript is not fluent at times. In my comments, I only focus on major flaws of the manuscript and skip my minor comments:

[Figure]

I noticed there are several unsupported claims in the manuscript, one of them is "parameter estimation". I can't find how the parameter estimation is performed. It is not close to sufficient to consider the 91 parameter combinations from the sensitivity analysis and use a selection criteria based on some fingerprints of the catchment to select parameters as behavioral from this limited set. This would lose many behavioral parameter combinations, and doesn't provide any information about the posterior distribution.

I might be wrong, but my understanding is that the sensitivity analysis of model parameters depends on a residual based objective function. At least it is highly dependent on simulation of the system response at individual time steps! This is in contrast with the purpose of using fingerprints of catchments, which are originally defined to constrain model parameters to represent an aggregate behavior of the catchment.

It is not clear how the selection criteria is adopted to delineate the behavioral parameter distribution. There are times that authors discuss one fingerprint is used, whereas in other instances they used a couple of fingerprints jointly! In the original application of fingerprints that authors referred to (Vrugt and Sadegh, 2013), 4 fingerprints were uses that are necessary to meet the acceptance criteria jointly. It is not clear if authors have performed their analysis on single sites (headwaters), or they have modeled the entire system altogether.

Page 2, line 32: My experience shows that, at least for US catchments, parameters of certain models are more correlated with climatic variables rather than soil characteristics. It is worth mentioning here, although the sentence is correct in how it describes the findings.

In section 3.3.2, authors talk about the study area before introducing it!

Page 6, Line 3: I don't understand the sentence: "relate physiographic and climatic characteristics to sensitivity-confined hydrodynamic response fingerprints"

Page 10, line 7: Water is withdrawn from the system, it is not lost!

[Figure]

Section 5.3: authors talk about consistency of behavioral parameter sets. In what sense have you analyzed the consistency of parameter sets? For definition of hydrologic consistency refer to: Martinez, G. F., and H. V. Gupta (2011), Hydrologic consistency as a basis for assessing complexity of monthly water balance models for the continental united states, Water Resources Research, 47 (12).

Page 15, line 25: In the entire manuscript authors are talking about 6 model parameters, and all of a sudden they switch to 52 global mHM parameters! It confuses me which one is the correct number of model parameters.

Page 16, line 26: Authors suddenly talk about temporal sensitivity of parameters! This is completely different from what reader expect from a joint sensitivity-parameter estimation analysis. The latter works with the entire data set, whereas the former is concerned about individual time steps!

Page 19 lines 6-16: Categorizing catchments based on model parameter assumes that the model is sufficiently describing the system. This assumption is not well justified nor supported by the results.

Page 20, lines 21-22: This is again unjustified claim to say this paper does: "(1) investigate hydrologically relevant structural and functional attributes in terms of consistency and feasibility in classifying similar catchments, (2) assess the value of functional constraints for the parameter spaces of distributed hydrologic models". I am not convinced that the results of this study support these claims.

Figure 3 is not well explained in the text.

Figure 10 & 11: How is it possible to differentiate between model simulations of the two gauges?

Figure 12: Why four of the parameters and not all 6?

---

## Author Comment (AC1) · 14 Aug 2016

Response to B. Guse (Referee)

**Björn Guse (BG)**: In this manuscript, Höllering et al. provide an approach to obtain a better understanding of model parameter behaviour and a more process-based model parameter estimation under consideration of spatial variations. For this, they used at first a fingerprint analysis to investigate how well the model performs for different aspects of the hydrological system. Second, a temporally resolved parameter sensitivity is used to detect the dominant model parameters along the time series. Overall, I really like the idea of this approach. However, I think that the core ideas of this study needs to be clarified. The introduction is not appropriate and needs to be reworked (including references). Furthermore, I think that the presentation of the results in the figures can be improved (also in their quality) and condensed, while the interpretation towards the overall benefit for the hydrological community can be enhanced. Thus, I recommend a major revision of this manuscript. For this, I make several recommendations below.

**Simon Höllering (SH)**: **We thank referee Björn Guse for his critical and helpful assessment of our manuscript.**

MAJOR COMMENTS

1. Introduction:

**BG:** The introduction needs to be completely reworked and restructured. I see here several reasons for this recommendation. The introduction is not related to the abstract. It is very surprising that the introduction starts with catchment classification / similarity after reading the abstract. Further, In the methods and results, the performance (fingerprint) analysis and the temporal parameter sensitivity analysis have at least the same priority. An introduction into the state-of-the-art in parameter sensitivity and more specifically on temporal parameter sensitivity analysis is completely missing. There are several studies in recent years using a temporally resolved sensitivity analysis and trying to extract helpful information for parameter understanding. Please see among others: Guse et al. (2014, 2016), Herman et al. (2013a, b), Massmann and Holzmann (2012), Massmann et al. (2014). Furthermore, I am not satisfied with the introduction into performance analysis in terms of fingerprint analysis and parameter constraints. Also here, there are several recent advances which need to be considered here to see this study in the context of the state-of-the art in research. Exemplarily please see: Euser et al. (2013, 2015), Gharari et al. (2014), Pfannerstill et al. (2014), Pokhrel et al. (2012), Reusser et al. (2009). Thus, it is certainly required to extract in a better way the novelity of this approach in the context of temporal parameter sensitivity analysis, performance analysis and constraints for model parameters compared to the state-of-the art.

**SH: We admit that the introduction should be streamlined to better reflect/lead to the key objectives of this study. One is certainly the question of what to learn from parameter sensitivity in different catchments, this is what we paraphrase as "regional sensitivity". The other is to shed light on the usefulness of catchment fingerprints for parameter identification. We think that the FAST method is well suited to address both objectives and we are happy to refer to the studies listed by Björn Guse, -in case they are relevant-**

2. Objectives:

**BG:** Following of my comments on the introduction, the objectives are not clearly enough motivated. Please check in the introduction whether all objectives are really motivated in the introduction. According to the current version of the introduction, I do not see a clear reason why it is relevant to constraint parameters in relation to different flow conditions or whether it is required to look at parameter sensitivity to understanding spatial distributed catchment behaviour. Certainly, I agree to both research questions, but I do not agree with their motivation.

**SH: We will rework the introduction and the related objectives, as stated above.**

3. Concept and methods:

**BG:** According to my understanding of the manuscript, the three pillars are not really representing the article. The major point seems to be the third part. The second point is not very specific (stream flow generation). I would recommend to emphasize here more the general idea by shortly explaining how the different parts are related and which benefit is intended to obtain by the different steps. This is currently a bit unclear. Especially when presenting the concept, a clear structure is required. Maybe a flowchart would be more helpful than a list of five steps.

**SH: We thank for this important point, the study is mostly focused on the two pillars - suitable indices to characterise stream flow generation as well as partial parameter sensitivities. We will clarify this part and possible add a flow chart.**

**BG:** The link between the introduction and the concept is also not clear. Catchment classification dominates the introduction, but is not included in four of the five research steps. The authors should think about directly considering the relationship between parameter sensitivity and dynamic fingerprints. This aspect is somehow missing at the end and could provide helpful insights into parameter understanding.

**SH: This is a great idea, we tried to achieve this with Figure 11, which relates the FDC and the highest parameter sensitivity for two different catchments. Highest sensitivities are quite different for both catchments. We will further explore this relation in the revised manuscript and elaborate on the related hydrological insights.**

4. Results:

**BG:** There is a huge amount of figures showing the results. My impression is that due to the content of the figures (and the different subplots), the overall goal of this study is somehow lost. I strongly recommend to focus on figures showing the major outcomes.

In the results (5.3), it is not clear which knowledge is really gained by using the best selected model runs for a certain fingerprint (Fig. 7a). I think that we are more interested in overall best performing model runs combining different indices than in having a model runs which is the best in relation to a single fingerprint. I had expected a presentation of a joined metric combining all fingerprints such as e.g. shown in Pfannerstill et al. (2014) or Haas et al. (2016), so that a model run could be finally selected which performs well for all fingerprints. In my opinion, this would be a reasonable final result of this part.

**SH: We will reduce the richness of figures to portray the main findings. A first guess on could at least hope, that parameter sets which reproduce well selected fingerprint would also perform well during a streamflow simulation. As this did not work out we selected parameter sets which perform well for at least 2 fingerprints, with moderate success. A number of 91 runs is surely too small to find parameter sets which work well for most of the fingerprints.... Next it might be interesting to select the parameters**

**in a different manner to represent different characteristics of stream flow (runoff generation parameters for magnitude, recession parameters for timing) as well as to optionally include a master recession curve as fingerprint or related recession indices. A joint metric to judge the performance of parameter sets with respect to different fingerprints is also an interesting option.**

**BG:** A discussion of the results in the context of the state-of-the-art and of how these results are related to knowledge obtained by former studies in this topic is missing. Concerning this, please see among others the list of references at the end.

I really like the expression Sensitivity duration curve (SDC). It is especially good to see that in this case differences between the catchments were detected. This is contrast to Guse et al. (2014) where a similar presentation (Fig. 6, even when it was not named Sensitivity duration curve) did not show relevant differences (due to a lack of spatial heterogeneity).

**SH: We will relate our findings to the those of recent studies recommended by BG and particularly refer to BG paper in the respect of sensitivity duration.**

**BG:** Maybe the authors could think about a final figure summarizing the results qualitatively. An example for this could be found in Fig. 9 in Herman et al. (2013a). Discussion and conclusion:

Concerning the stated research questions, I think that the first research question is not really solved. I agree that fingerprints can help in constraining parameter ranges. However, this was also expected, since each hydrologic metric can somehow constraint parameter range. Here, I miss either a method to select overall behavioural parameter sets based on all (or all not correlated ones) fingerprints or a hydrological explanation that a certain fingerprint is able to constrain a parameter range since it represents the associated parameter accurately or something similar.

**SH: Good point, we will find a way to better summarize overall results and discuss the value of overall behavioural parameter sets in terms of constraining the parameter space if the whole set of response fingerprints is considered.**

The sentence on P. 18, L. 26-27 "We further found..." really makes a strong difficulty apparent. The results that the parameter values (or constraints) are largely varying between the different behavioural parameter sets are problematic since it is then difficult to estimate the "best" parameter values. I encourage the author to discuss this point more in detail by suggesting possible way towards an overall behavioural parameter set.

**SH: An alternative interpretation is that these changes show either that the sample is too small or they suggest that the model structurally inadequate. A perfect model should yield parameter sets which consistently reproduce at least related signatures.... if this is not the case, although the model yields a good NSE, this might be due to structural inadequacy of the model.**

**BG:** In this context, I think that a more profound discussion of the relationship between the performance of fingerprints and model parameters would be helpful. At the end of the answer to research question 2 (P. 19, L. 6-16), the consistency between sensitivity duration curves and hydrologic fingerprints can be discussed. Do the spatial patterns of both are consistent?

**SH: We will improve the discussion in the revised manuscript along these lines.**

5. Figures:

**BG:** All figures: Overall, please check which are the most important figures and which are of less importance and could be removed/reduced.

In relation to this, several figures are not clear enough in terms of their intention and the visibility of the results. In particular, the description of the results in the figures (e.g. Figs. 5 and 7) is sometimes difficult to grasp. Concerning this remark, please see also the following comments to the figures.

**SH: We will reduce the number of figures to the minimum necessary amount.**

**BG:** Fig. 5: This figure needs to be improved. It is impossible to extract the information of the relationship between same coloured points and the circle. One idea could be a reduction of the selected stations (in this plot) and/or a quadratic plot (increase of the plot height). Another idea could be to add a table (maybe as acknowledgement) stating how many points are within the circles. To summarize this comment: It should be possible to extract the information of how many points are in a circle somehow.

**SH: Good point, we will find an appropriate presentation for this.**

**BG:** Fig. 6: I do not understand why and how the sample space is defined by the observed quantities in a fingerprint. Here, a clear approach is missing. I can agree that in the best case all simulations should be in the range as defined by the observed values of all stations. However, how is this related to the parameter space? This requires at least a detection of the parameter values leading to the fingerprint values as well as a clear relationship between parameter value and fingerprint. Maybe I understand something wrong here, but for me it seems to be that an information is missing here. It is really crucial for this study to understand this point.

**SH: This is a misunderstanding. Figure 6 shall corroborate that the fingerprints of the fast ensembles spread across different ranges for the different catchments. As the variation of parameters was the same, this indicated regional how differences in forcing data, propagate into FAST. Second the plot shows that the fingerprints differ for the different catchments, i.e. they are sensitive to regional differences and third it shows that the envelopes are highly skewed.**

**BG:** Fig. 7a: This sub-figure needs to be completely reworked and improved. I cannot extract the relevant information. There is too much information: Seven sites as colours, six fingerprints as symbols and two metrics as well as the number of the best performing model run. One idea could be a plot in similar way as Figs. 1-6 in Bastidas et al. (2006). In this case, a separate plot could be shown for each performance metric.

**SH: We will rework this figure to better portray the main finding. We agree that there is too much information in it.**

6. MINOR COMMENTS:

**BG:** Abstract, first sentence: I think that the first sentence should be more general. It is not clearly apparent why parameters need to be identified in relation to parameter sensitivity and catchment classification.

**SH: Yes, we are thinking about a more general/clear formulation.**

**BG:** Abstract, second sentence: This sentence is certainly too long. Please subdivide this sentence into two (or even three) to avoid losing the reader directly at the beginning.

**BG:** Abstract: The numbering (1), (2) and (3) is not explained and thus not understandable.

**SH: We split the long sentence and removed the numbering.**

**BG:** The referencing should be consistent. To give an example: On Page 5, Line 25, the three references are neither ordered by occurrence nor alphabetic. Please check this in the whole text.

**SH: Yes, thanks, corrected.**

**BG:** At several parts of the manuscript, the transitions between the different subchapters are not clear. I recommend to check the beginning and the end of the different chapter and if required add a sentence to relate both chapters. One example for this is the beginning of chapter 5.4.

**SH: We will incorporate transitions to improve readability where it is needed.**

**BG:** P. 7 L. 3: Why do you use only six parameters which only explain less than the half of the variance?

**SH: The selection of six parameters originated from a pre-study, were a local sensitivity analysis was performed to find out sensitive parameters in terms of different indices for streamflow dynamics and water balance. 14 parameters were identified as most sensitive in this way and used for a pre-run of FAST whereby the six most influential parameters where kept to demonstrate the feasibility of our approach with a comparably small amount of model runs/time to be spent on simulation. We think about restructuring the parameter pre-selection for future publication.**

Later in the text (P. 15, L. 24), it is mentioned that even on the day with the highest sum of the partial sensitivities, this value is lower than 0.5. I think that a good reasoning for this is required. Even though that I am aware that it is not useful to do a temporal sensitivity analysis with all model parameters, I am curious whether it is possible to increase the explaining variance by using e.g. 8 or 10 parameters. Or otherwise, could you explain why a higher number of parameters is not beneficial/possible?

**SH: Yes, this might be demonstrated later in a different study, where more parameters might be included as well.**

**BG:** P. 7, L.11: I would recommend to write here: "In the case of using six parameters, the FAST method requires altogether 91 model runs..." It should be highlighted that the number of model runs depends on the model parameters and is provided by the FAST methods meaning that the same number of model runs is required for the same number of parameters independently from model, catchment or parameter selection.

**SH: Yes, we corrected it and will further clarify this part.**

**BG:** P. 7, L.25: Here, 91 model runs are used to identify values for six parameters. This approach is certainly in contrast to typical model calibration algorithms using a significantly higher number of model runs for the same number of parameters. Even when I agree of using this approach for this study, I think that it is required to mention that a lower number of model runs is acceptable here according to capture all goals of this study. Or other way round, it is required to say that this number of model runs has to be certainly higher when only focusing on model optimization.

**SH: Good point! We will consider this in the revised manuscript.**

**BG:** P. 8, L. 6: Which are the five classes?

**SH: We added a reference to table 2 showing the five classes.**

**BG:** P. 8, L. 17: There are more than six hydrologic fingerprints in Table 2. Could you explain why you mentioned here "six" hydrologic fingerprints?

**SH: This is a misunderstanding. There are six fingerprints (including BFI) which characterize the dynamic response of catchments (hydrological). The other part of the table shows the physiographic characteristics. We will rework this table to avoid misunderstanding.**

**BG:** P. 8, L.19: Which one (of the dynamic fingerprint)?

**SH: We change it to be better understandable.**

**BG:** P. 8, L. 26: I would recommend to write: "from each of the simulated..."

**SH: Yes, corrected.**

**BG:** P. 8, L. 30: I think that here and maybe also in the introduction a discussion of the PAWN method is missing as proposed by Pianosi and Wagener (2015) since the role of different performing model results within the sensitivity analysis is directly included in PAWN.

**SH: Yes, this seems to be interesting in the context of using global sensitivity analysis. We will try to consider this approach in the introduction part.**

**BG:** P. 9, L.-10-25: I recommend to also refer here to the work from Pfannerstill et al. (2015) and Pokhrel et al. (2012). In both studies FDC and their segments are used to identify (constrain) parameter values.

**SH: Yes, thank you for the hint.**

**BG:** P. 11, L. 12-13 (Fig. 5): Could you explain why you used circles assuming that the variation are similar for both variables. Is it maybe more useful to use an ellipse (which it would be in the case of a quadratic plot)?

**SH: As a qualitative constraint a circle defined by its radius was in our opinion enough to show the feasibility of the approach and capture its goals adequately. Nevertheless, an ellipse might be an even better choice.**

**BG:** P. 11, L. 12-16: I did not understand how the radius is selected and why it has this size.

**SH: Please also refer to the previous point. For each radius a scaling factor is introduced which can be regarded as a measure of distance orientated towards the overall data spread of the 14 derived 15 values of one fingerprint. The factor is determined as a multiple n of the average of the two standard deviations of the pairs of fingerprints and shows up to be a feasible graphical measure to distinguish behavioural from non-behavioural sets for a pair of fingerprints.**

**BG:** P. 11, L. 21: Could you explain why you mentioned both distance measures? Is it maybe more appropriate to select one (the best) of them?

**SH: Yes, we should focus on one them.**

**BG:** P. 12, L. 21-28: Which result (figure) supports this text passage?

**SH: There was no figure submitted which shows the result for the BFI. We will add it or remove this part in the revised manuscript.**

**BG:** P. 13, L. 20: Please explain why you have selected these seven gauging stations

**SH: We will add an explanation why we selected a subset of seven gauges.**

**BG:** P. 14, L. 2: Why do you selected four fingerprints and calculate an Euclidean distance between them. Why not using all fingerprints?

**SH: As stated earlier a joint metric to judge the performance of parameter sets with respect to all fingerprints is also an interesting option and might be used in a revised version.**

**BG:** P. 14: I have expected a clearer description of the intention of each subplot of Fig. 7 and a presentation of the major outcome of each subplot.

**SH: We will rework the presentation of Fig. 7 and it outcome.**

**BG:** P.16, L. 26: Please add here or later in the discussion that the relationship of parameter sensitivities and FDC (or sorted discharge) was already captured e.g. in Herman et al. (2013b) and Guse et al. (2016) if not already included in the introduction after revision.

**SH: Yes, this might be important here or better earlier in the introduction.**

**BG:** P. 16, L. 29: Is there a reason why two observed and only one simulated station are used in Fig. 11?

**SH: This is for the reason of legibility and due to poor model performance of one of the two stations.**

**BG:** P. 20, L. 21: I would recommend to structure the discussion in two sub-chapters to avoid a misunderstanding evoked by a double-use of the numbers 1-3 in the discussion. The second sub-chapter in the discussion could start at this line.

**SH: This is a good idea.**

**BG:** P. 21, L. 12: Why not directly increasing the number of parameters in this study?

**SH: As stated above and in the manuscript the number of parameters and the parameter itself were preselected in a previously carried out local sensitivity study. We regard the number of parameter as sufficient to test the feasibility of our approach and the selection makes sense in respect to the fingerprints we use here. A higher number of parameters would also considerably increase the number of necessary model runs, and with this compromise a key advantage of FAST. We agree that the additional variation of parameters of for instance a recession parameter could provide additional insight, but not necessarily for the selected fingerprints.**

**BG:** Fig. 1: I do not see the relationship between the FAST sampling design the parameter values in the calibration. Why do you show both in one plot? Which information can be derive from this relationship? Furthermore, due to the different ranges of the parameters, the interpretation of the parameter values is rather difficult.

**SH: We agree with the reviewer and will remove the calibrated parameters from the revised manuscript.**

**BG:** Fig. 1, caption: Please changed to "parameter values in the 91 model runs according to the FAST sampling" or a similar expression.

**SH: Yes, changed.**

**BG:** Fig. 2: Please increase the labels a and b in the figure.

**SH: Yes, changed.**

**BG:** Fig. 2: I strongly recommend to subdivide this figure into two plots showing separately hydrologic fingerprints and the dynamics response.

**SH: This might be a good point to consider for the revision.**

**BG:** Fig. 2b: The lines in the dynamic response fingerprints are unclear.

**SH: An explanation of the lines will be added**

**BG:** Fig. 3: Do you really need this figure? I do not see the real benefit.

Fig. 3: Yilmaz et al. (2008) made a FDC segmentation at 20% and not at 30% of flow exceedance.

Fig. 3: Are you showing here the 91 model runs as FDC? In this case it would suggest to clarify this by stating this.

**SH: We will probably remove this in the revision of the manuscript or further clarify your points.**

**BG:** Fig. 4: Since the gauges and their abbreviation are used several times in the manuscript, I strongly recommend to increase the labels in size. Maybe a white background (for the labels) would be helpful in addition.

**SH: Yes, this is certainly helpful.**

**BG:** Fig. 7: Please think about the benefit of each subplot.

Fig. 7: The legend to the gauges belongs to Fig. 7a and not 7f.

Fig. 7b: Please discuss in the text why the best performing run in relation to SLFDC is among the worst runs related to NSE.

**SH: Fig. 7 will be optimized and its discussion clarified.**

**BG:** Fig. 8: Please add in the figure caption that the numbers in brackets in the legend are the numbers of the model runs.

**SH: Added.**

**BG:** Fig. 10: Please explain in a better way: "highest parameter sensitivity related observed hydrograph".

**SH: Changed.**

**BG:** Fig. 11: It seems to be that the major information from these plots could be extracted in a simpler way. I do not think that the grey lines are required. What about showing only the changes in the dominant parameters as a line (or a row) for each gauge.

**SH: Yes this might be a good and interesting idea to optimize this figure.**

**BG:** Fig. 12: Maybe the legend could be shown only once and outside of the plot at the right side (only a very minor comment).

**SH: Yes, we also think about showing SDCs of all the six parameters.**

7. Technical corrections:

**BG:** P.1, L. 10: sensitivity

**BG:** P.4, L. 16-18: This sentence does not read well.

**BG:** P. 5, L.6: I recommend to use the paper of Reusser et al. (2011) instead of the dissertation work (Reusser 2010).

**SH: Thank you, we corrected all of the technical shortcomings you stated.**

8. References:

**BG:**

Bastidas LA, Hogue TS, Sorooshian S, Gupta HV and Shuttleworth WJ (2006): Parameter sensitivity analysis for different complexity land surface models using multicriteria methods, J. Geophys. Res., Vol.11, D20101, doi:10.1029/2005JD006377.

Euser T, Winsemius HC, Hrachowitz M, Fenicia F, Uhlenbrook S and Savenije HHG (2013): A framework to assess the realism of model structures using hydrological signatures, Hydrol. Earth Syst. Sci, 17(5), 1893-1912.

Euser T, Hrachowitz M., Winsemius HC and Savenije HHG (2015): The effect of forcing and landscape distribution on performance and consistency of model structures, Hydrol. Process. 29(17), 3727-3743.

Gharari S, Shafiei M, Hrachowitz M, Kumar R, Fenicia F, Gupta HV and Savenije HHG (2014): A constraint-based search algorithm for parameter identification of environmental models, Hydrol. Earth Syst. Sci., 18, 4861-4870.

Guse B, Reusser DE and Fohrer N (2014): How to improve the representation of hydrological processes in SWAT for a lowland catchment – temporal analysis of parameter sensitivity and model performance. Hydrol. Process. 28: 2651–2670.

Guse B, Pfannerstill M, Strauch M, Reusser D, Lüdtke S, Volk M, Gupta H and Fohrer N (2016): On characterizing the temporal dominance patterns of model parameters and processes, Hydrol. Process., 30(13), 2255-2270.

Haas M, Guse B, Pfannerstill M and Fohrer N (2016): A joined multi-metric calibration of river discharge and nitrate loads with different performance measures, J. Hydrol., 536, 534-545. Herman JD, Kollat JB, Reed PM and Wagener T (2013a): From maps to movies: high resolution time-varying sensitivity analysis for spatially distributed watershed models. Hydrol. Earth Syst. Sci. 17: 5109–5125.

Herman JD, Reed PM and Wagener T (2013b): Time-varying sensitivity analysis clarifies the effects of watershed model formulation on model behavior. Water Resour. Res. 49. DOI:10.1002/wrcr.20124.

Massmann C and Holzmann H (2012): Analysis of the behavior of a rainfall–runoff model using three global sensitivity analysis methods evaluated at different temporal scales. J. Hydrol. 475: 97–110.

Massmann C, Wagener T and Holzmann H (2014): A new approach to visualizing time-varying sensitivity indices for environmental model diagnostics across evaluation timescales. Environ. Model. Softw. 51: 190–194.

Pfannerstill M, Guse B and Fohrer N (2014a): Smart low flow signature metrics for an improved overall performance evaluation of hydrological models. J. Hydrol. 510: 447–458.

Pianosi F and Wagener T (2015): A simple and efficient method for global sensitivity analysis based on cumulative distribution functions, Environ Model Softw. 67, 1-11.

Pokhrel P, Yilmaz KK and Gupta HV (2012): Multiple-criteria calibration of a distributed watershed model using spatial regularization and response signatures. J. Hydrol. 418: 49–60.

Reusser DE, Blume T, Schae i B and Zehe E (2009): Analysing the temporal dynamics of model performance for hydrological models. Hydrol. Earth Syst.Sci. 13: 999–1018. Interactive comment on Hydrol. Earth Syst. Sci. Discuss., doi:10.5194/hess-2016-249, 2016.

**SH: Thank you for this reference list. We will add citations to these references.**

---

## Author Comment (AC2) · 16 Aug 2016

Response to S. Gharari (Referee)

**Shervan Gharari (SG)**: After reading the manuscript, I must admit that "I am lost and confused"! It seems the manuscript is revolving around many but at the same time no clear message. Either I missed or I am not capable to fully understand; I don't see any conclusion from the presented manuscript! Starting from the structure; the manuscript is very badly structured and too wordy and long. The literature review is spread over the paper and there are a lot of unnecessary sub-sections and subsub-sections which can be reduced. The title looks very sophisticated and broad. Every word need further explanation which the manuscript fails to fully explain. As an example, "integrated" and "sensitivity-nested" seem to have similar meaning. "Multi-fingerprint" can be replaced by much simpler words. Did the authors really carry out a "catchment similarity assessment" as stated in the title? The abstract is again distance from what the paper is trying to tell. One cannot really understand the final conclusion of the paper from the abstract and only an unclear and again broad sentence such as "The sensitivity approach may be useful. . ." concludes the abstract. I would also remove the world "novel" from the abstract and any other places in the manuscript. It is the readers' decision to decide about the novelty not the authors'.

**Simon Höllering (SH): We thank Shervan Gharari for his critical assessment of our manuscript. We will reword the title and stream line the introduction, as outlined in our response to Björn Guse.**

**SG:** The introduction is very wide too with many relevant and irrelevant studies put together next to each other in non-coherent sentences. The general literature review is not enough and missing a significant body of the existing literature on catchment similarities, catchment classification and so on. As an example, one of my own research interests which have been mentioned in section 1.2 can be re-written for topographical landscape unit as follow: "Using indices based on topography can be one of many ways to delineate a catchment into hydrological response units or as stated by Zehe et al., 2014, functional units. These units can be built based on topographical features as stated by Knudsen et al., 1987, Flugel 1995 and winter 2001 into X1, X2 and X3. Aligned with what have been suggested and with help of a recent topographical index (HAND, Renno et al., 2008) Gharari et al (2011) classified a small (or meso scale) catchment in Luxembourg and then used the mapping for building and constraining a conceptual model (Gharari et al 2014a). This classification have been repeated by Gao et al 2014 for a large scale catchment in China and served as the basis of the modeling exercise."

**SH: We thank SG for this statement and will revise the passage accordingly.**

**SG:** The authors can extend this section by elaborating on slope, soil and land cover. Please keep in mind that the introduction, as well as each paragraph, should have a funnel shape structure starting from broad and general overview and ending in example of specific implications. This ways the reader will be ready for the final and general message of the paper.

**SH: We thank SG also for this statement and admit that the introduction and some other sections should be streamlined.**

**SG:** "Is the inconsistency of functional unit and physiographic similarities a paradox?" No one really claim that response units should exactly follow physiography. This is just an assumption to give us the ability to make of prediction of what we really don't know (such as prediction in ungauged basin).

**SH: We thank SG for this comment, but we have a slightly different view. Hydrological processes are essentially determinist, statistics comes into play to deal with non-exhaustive observations. Hence, one would expect that structurally similar catchments produce a similar response behavior, when being subject to similar forcing. The fact that up to now structural similarity and functional similarity are often not consistent, might be explained by many factors (non- exhaustive observations, separated treatment of structural characteristics, which jointly control runoff (such as soil and topography) and many more ....**

**SG:** I encourage the authors to give a comprehensive literature review in section 1.4 regarding the signatures (or fingerprints). I would also ask the authors to give a comprehensive literature review on the use of signature in the hydrological modeling (such as Euser et al., 2013 and Clark e t al., 2011).

**SH: As also recommended by the first reviewer we will better focus on fingerprints used in hydrological modeling and e.g. as constraints for the parameter space and in model diagnostics. We will rework the literature review on this to be more comprehensive.**

**SG:** What remains missing in the introduction given the gist of title and current introduction is the relation between the model, signature and processes. There have been numerous studies looking on this topic, the sensitivity and also uncertainty of the model parameters over different period of time. I remember reviewing a manuscript on this topic for the very same journal HESS, Pfannerstill et al., 2015. Dr. Guse gave a very wide range of publication related to this in his review.

**SH: Yes, we will clarify the relations and try to better focus on (temporal) parameter sensitivity related to fingerprints characterizing hydrological processes and consistency in model performance.**

**SG:** Section 3.4.2 is a mixture of literature review of the signatures, and FAST and selected signature for this study. I would advise the authors to briefly clarify what signatures they used in this study. The literature review should have been presented earlier and FAST is explained later in the manuscript which makes it a bit difficult to follow. I would also suggest to clarify the possible limitation of FAST compare to the other sensitivity analysis, Razavi and Gupta (2015) may help in that regard.

**SH: We will rework this section and clarify the limitations of the FAST approach, for instance it's insufficiency to deal with parameter interdependence.**

**SG:** Please clarify the name and possibly the similarities of the headwater catchments in section 4.1 and 4.2. The names of the headwaters pop up every now and then in the Discussion paper manuscript!

**SH: Yes, we will make this point more understandable in these sections.**

**SG:** About the parameter constraining: In my point of view what the authors are presenting in this study is not parameter constraining, it is rather parameter selection given various signature and based on sensitivity analysis. We had recent papers looking at constraining (Gharari et al., 2014a and 2014b, Hrachowitz et al., 2014). There are many related existing studies as well, the authors are more than welcome to address in them in their manuscript. Constraining should be different in my point of view from parameter selection based on a multi objective approach (which is similar to breaking the evaluation of the time series into different signatures).

**SH: We will clarify this section. Although we regard the use of fingerprints to act as a constraint on the feasible parameter space. This is corroborated by the findings in the paper, because the best parameter set in terms of the NSE does not perform well with respect to the signatures. We are aware of the work of Hrachowitz, which for among other constraints us the position on the Budyko curve to constrain the long term water balance. At least this is similar to our approach. However, we do constrain parameters in the sense that the recession parameters of the slow flow reservoirs to be "smaller" than of the faster reservoirs.**

**SG:** The figures and result part is very hard to follow. I believe one of the main reasons is the simultaneous presentation of the results together with the methods. I would have separated the method and result sections. The figures are almost very difficult to follow, as an example figure 5, 6, 7, 8, and 11.

**SH: We will streamline the presentation of the results and their discussion. We will rework figures to better portray the main findings. We agree that in some of them there is too much information, which should be further clarified. We further want to reduce the number of figures to the minimum necessary amount with relevant information.**

**SG:** Again back to what I mentioned earlier the authors are not constraining the parameter sets but they are selecting different parameter sets based on different criteria (Figure 7). The criteria the authors are using to me seems like a multi-objective approach where the most balanced parameter sets are selected by Euclidian distance. We did this in our study in 2013 as well (Gharari et al, 2013) using Pareto front members and Euclidian distance to pick the behavioral parameter sets. The point of that study was not to constrain the model parameters but to show that the parameter sets which are better performing over time are different from the optimal or behavioral parameter sets.

**SH: We agree with your point of view but also please refer to the answer stated earlier. We regard the selection of several parameter sets/combinations by observed fingerprints as constraints on the feasible parameter space. We can additionally further characterize the constraint parameter space by analyzing behavioural parameter sets in terms of their pairwise distances in the 6-d vector space by the Chebyshev distances and can describe them in relation to the entire parameter space which is in this way constrained by the behavioural sets.**

**SG:** Conclusion and discussion is vague as well. I would suggest the authors to separate the conclusion and discussion parts. If the paper have a message and strong conclusion the conclusion part should be only few bullet points, pointing at the most general and specific findings of the manuscript. The authors can make use of this bullet point conclusions to form the abstract, introduction and methodology better.

**SH: Yes, two other referees also recommended to split this section into two sub-chapters. We will incorporate this in the revised version of our manuscript.**

The manuscript is far from the minimum quality for publication, meaning that it should be rejected. To give a chance for the manuscript contribution to be published in HESS I would give major revision. I am looking forward to receiving the revised manuscript. I hope my comments help the authors to elevate the level of the current manuscript and present in much better shape.

With kind regards

Shervan Gharari

PS. I thank Fuad Yassin who helped me better understand the manuscript. Our discussion was helpful for writing this review. I guess he was also "lost and confused"! ;)

References:

Gharari, S., Hrachowitz, M., Fenicia, F., and Savenije, H. H. G.: An approach to identify time consistent model parameters: sub-period calibration, Hydrol. Earth Syst. Sci., 17, 149–161, doi:10.5194/hess-17-149-2013, 2013.

Gharari, S., Hrachowitz, M., Fenicia, F., Gao, H., and Savenije, H. H. G.: Using expert knowledge to increase realism in environmental system models can dramatically reduce the need for calibration, Hydrol. Earth Syst. Sci., 18, 4839-4859, doi:10.5194/hess-18-4839-2014, 2014.

Gharari, S., Shafiei, M., Hrachowitz, M., Kumar, R., Fenicia, F., Gupta, H. V., and Savenije, H. H. G.: A constraint-based search algorithm for parameter identification of environmental models, Hydrol. Earth Syst. Sci., 18, 4861-4870, doi:10.5194/hess-18-4861-2014, 2014.

Gharari, S., Hrachowitz, M., Fenicia, F., and Savenije, H. H. G.: Hydrological landscape classification: investigating the performance of HAND based landscape classifications in a central European meso-scale catchment, Hydrol. Earth Syst. Sci., 15, 3275-3291, doi:10.5194/hess-15-3275-2011, 2011.

Euser, T., Winsemius, H. C., Hrachowitz, M., Fenicia, F., Uhlenbrook, S., and Savenije, H. H. G.: A framework to assess the realism of model structures using hydrological signatures, Hydrol. Ear th Syst. Sci., 17, 1893-1912, doi:10.5194/hess-17-1893-2013, 2013.

Hrachowitz, M., Fovet, O., Ruiz, L., Euser, T., Gharari, S., Nijzink, R., Freer, J., Savenije, H. H. G., and Gascuel-Odoux, C.: Process consistency in models: The Interactive importance of system signatures, expert knowledge, and process complexity, Water Resour. Res., 50, 7445–7469, doi:10.1002/2014WR015484, 2014

Gao, H., Hrachowitz, M., Fenicia, F., Gharari, S., and Savenije, H. H. G.: Testing the realism of a topography-driven model (FLEXTopo) in the nested catchments of the Upper Heihe, China, Hydrol. Earth Syst. Sci., 18, 1895–1915, doi:10.5194/hess-18-1895- 2014, 2014.

Zehe, E., Ehret, U., Pfister, L., Blume, T., Schröder, B., Westhoff, M., Jackisch, C., Schymanski, S. J., Weiler, M., Schulz, K., Allroggen, N., Tronicke, J., van Schaik, L., Dietrich, P., Scherer, U., Eccard, J., Wulfmeyer, V., and Kleidon, A.: HESS Opinions: From response units to functional units: a thermodynamic reinterpretation of the HRU concept to link spatial

organization and functioning of intermediate scale catchments, Hydrol. Earth Syst. Sci., 18, 4635-4655, doi:10.5194/hess-18-4635-2014, 2014.

Flügel, W.-A.: Delineating hydrological response units by geographical information system analyses for regional hydrological modelling using PRMS/MMS in the drainage basin of the River Bröl, Germany, Hydrol. Process., 9, 423–436, doi:10.1002/hyp.3360090313, 1995.

Rennó, C. D., Nobre, A. D., Cuartas, L. A., Soares, J. V., Hodnett, M. G., Tomasella, J., and Waterloo, M. J.: HAND, a new terrain descriptor using SRTM-DEM: Mapping terra-firme rainforest environments in Amazonia, Remote Sens. Environ., 112, 3469–3481, doi:10.1016/j.rse.2008.03.018, 2008.

Martyn P Clark, Hilary K McMillan, Daniel BG Collins, Dmitri Kavetski, Ross A Woods, Hydrological field data from a modeller's perspective: Part 2: process A Rbased evaluation of model hypotheses, Hydrological Processes, 25, 4, 523-543, 2011

Fenicia, F., D. Kavetski, H. H. G. Savenije, and L. Pfister, From spatially variable streamflow to distributed hydrological models: Analysis of key modeling decisions, Water Resources Research(52), 1-36, doi:10.1002/2015WR017398, 2016.

Pfannerstill, M., Guse, B., Reusser, D., and Fohrer, N.: Process verification of a hydrological model using a temporal parameter sensitivity analysis, Hydrol. Earth Syst. Sci.,19, 4365-4376, doi:10.5194/hess-19-4365-2015, 2015.

Razavi, Saman, and Hoshin V. Gupta. "What do we mean by sensitivity analysis? The need for comprehensive characterization of "global" sensitivity in Earth and Environmental systems models." Water Resources Research 51.5, 3070-3092, 2015.

Knudsen, J., Thomsen, A., and Refsgaard, J. C.: WATBAL A SemiDistributed, Physically Based Hydrological Modelling System, Nord. Hydrol., 17, 347–362, 1986.

Winter, T. C.: The Concept OF Hydrologic Landscapes, J. Am. Water Resour. Assoc., 37, 335–349, doi:10.1111/j.1752-1688.2001.tb00973.x, 2001.

**SH: Thank you for this reference list. We will consider it in the revised manuscript.**

---

## Author Comment (AC3) · 16 Aug 2016

Response to F. Sarrazin (Referee)

**Fanny Sarrazin (FS)**: In this manuscript, the authors present a parameter estimation and sensitivity analysis scheme to learn about differences in catchment functioning and to classify catchments. However, many points need clarification, in particular the implications and conclusions of the work. I recommend major revisions of the manuscript. Parts of the manuscript appear to be quite long (specifically the introduction and results sections) and the reader tends to get lost. A general recommendation to the authors is to select in each section the key elements that contribute to their argumentation and to clearly connect them. The authors should also try to keep their sentences short to improve readability. Furthermore, critical information on the implementation of sensitivity analysis is missing (output considered, how the sample size was chosen), which makes it difficult to interpret the sensitivity analysis results. I provide below more detailed comments for the different sections and some minor comments at the end.

**Simon Höllering (SH): We sincerely thank Fanny Sarrazin for the time and efforts she invested into the review of our manuscript. We agree that the presentation of our study can be considerably improved and will do this in the revised manuscript as specified in the forthcoming reply to the reviewer comments.**

SECTION 1

Introduction

**FS:** 1) I think the introduction section is unnecessarily long and not sufficiently linked to the objective section (section 2). Specifically, how do the work of Bardóssy (2006) (p2 L11) Wagener et al. (2007) (p2 L18), Castiglioni et al. (2010) (p4 L14), Yadav et al. (2007) (p4 L18) relate to the work presented in the manuscript? In which way the literature review on catchment classification (section 1.2) is useful to better understand the work presented in the manuscript? I suggest the authors select key elements in the literature review and clearly link them to their study, so to clarify what the contributions of the manuscript are.

**SH: We will streamline the introduction and focus on studies dealing with regional parameter sensitivity, FAST and their optional combination. As also recommended by the first reviewer we will better focus on fingerprints used as constraints for the parameter space and model diagnostics.**

**FS 2):** There is some confusion in Section 1.3 and it needs clarification. A distinction has to be made here between a model and the underlying system it represents. The term equifinality commonly refers to the model and not the system it represents. In fact, Beven (2006) (p21, paper cited by the authors) characterizes the term equifinality as the 'rejection of the assumption that a single correct representation of the system can be found given the normal limitations of characterisation data.' Therefore, equifinality is not an 'inherent' property of a model equations as stated by the author p3 L31, but it is due to the fact that the information content in the data is not sufficient to identify a unique model representation. I am also asking the question: Is the study of physical and hydrological similarities based on observation data or on model simulation outcomes?

**SH: We thank for this comment, which shows the need for clarification. We agree that the system is not equifinal, hence the average residence time of water in a catchment is well defined (might be state dependent). What we meant with equifinality is inherent to our equations is equifinality due to parameter interactions. For instance n and k determine the average "travel time" through a Nash Cascade. Bardossy (2007) showed nicely that an infinite number of parameter yield acceptable results – a doubled n in combination with a 50% reduce k yields the same results. The same holds true for**

**Darcies law, as system with a twice as large gradient and 50% reduced permeability produces the same flux. We will better explain this in the revised manuscript**

**In the revised manuscript we will also explain that we use observed streamflow signatures, which have been shown to vary among our target catchments, for regional model evaluation. On one hand we test whether parameter sets which reproduce one or multiple fingerprints, yield acceptable stream flow predictions (or the other way around). A structurally adequate model should be consistent with respect to both. We furthermore use FAST to estimate the sensitivity of the fingerprint of the parameter along their independent variable (for instance for different exceedance probabilities when using the flow duration curve. Differences among different catchments might be a hint for different interplay of runoff generation processes. This information might be furthermore useful for calibrating models directly based on fingerprints.**

**FS:** 3) p4, L20: the expression 'to raise the information considered in the parameter transfer' is quite fuzzy. Please clarify.

**SH: We apologize for being unprecise. We will find a more comprehensible formulation for the reference to these studies in the revised version of this section.**

SECTION 2 Objective and driving research questions

**FS:** 1) As mentioned above, the objectives need to be clearly linked to the literature review, Interactive to show the contributions of the manuscript.

**SH: Yes, we will link the review more precisely to what we want to achieve with our study. Clearer objectives could be:**

- **Explore regional differences in TEDPAS (Temporal Dependence of PArameter Sensitivity) and discuss their implications for model diagnostics.**
- **Extend the TEDPAS concept to fingerprints to explore and depict changes in their parameter sensitivity and along independent variables (e.g. exceedance probability for FDCs).**
- **Employ constraints by fingerprints on the feasible parameter space of a distributed hydrologic model and assess hydrological consistency in terms of model performance and model structural aspects.**

**FS:** 2) p4 L30-31: This objective needs clarification. First, 'consistent manner' is vague and it is required to better explain this expression. Second, the motivation needs to be clarified. Is the objective to learn about the model ability to reproduce the data, to learn about the value of observation data etc.?

**SH: Sorry for being not precise here. We think that a structurally adequate model with behavioral parameters sets should consistently allow for acceptable stream flow simulations and reproduce fingerprints derived from stream flow. An inconsistent behavior in the sense that parameters which work well during stream flow simulations, but do not work well with respect to reproduce fingerprints provides evidence for model structural error. In this sense we tested here parameter sets which performed well with respect to one or several stream flow indices with respect to their performance during a stream flow simulation and vice versa.**

**FS:** 3) p5 L8-9: the expression 'within the range of their independent variable' needs clarification.

**SH: Sorry for being imprecise. We use the Fourier Amplitude Sensitivity Test to assess temporal dependence of parameter sensitivity. We hence vary 6 parameters with preselected independent frequencies within 91 model runs. The idea of FAST is then to perform a Fourier analysis of the 91 model runs for each time step. The amplitudes in the power spectrum provide information which of the parameter contributes most, second most and so on to the spread of the model runs at a given time step. Temporal changes in their weights indicate changes in dominant sensitivities. We do the same for stream flow signatures such, in case they are in the form of a bivariate plot such as the flow duration curve or a double mass curve of a catchment, which is accumulated discharge plotted against accumulated precipitation. The fast ensemble yields 91 flow duration curves or 91 double mass curves. Again we can perform a Fourier analysis of the 91 model runs for a given value of the abscissa (the independent variable). The amplitudes in the power spectrum provide information which of the parameter contributes most to the spread of the model runs. Changes in the weights of the different parameters indicate changes in dominant parameter sensitivities of the fingerprint either for different exceedance properties or for different accumulated rainfall inputs during a hydrological year.**

SECTION 3 Concept and methodological approach

**FS:** 1) I have reservations regarding the implementation of the sensitivity analysis (section

3.3), specifically:

- p7 L12: How was the sample size chosen? It is necessary to check the convergence of the results, that is to say to assess to what extent the results would change if using a new sample. When the sample size is too small, the sensitivity analysis results can be unreliable (e.g. Sarrazin et al., 2016). A way to assess convergence is to derive confidence intervals on the sensitivity indices using bootstrapping. However, this technique cannot be easily applied when using FAST, given the structure of the sample. Therefore, I would suggest to simply look at the stability of the results when using smaller/larger sample sizes.

**SH: As far as we understand the method, and as explained in Reusser et al. 2011, the frequencies and the minimum number of model runs are pre-defined by the theory of the FAST Fourier transform method. The sampling size depends on the number of parameters and independent frequencies. We hence do not expect different results when increasing the number of model runs. Results might however definitely change when the sensitivity of the model is heteroskedastic, for instance in the presence of threshold processes, for instance Hortonian overland flow. When choosing a specific range for the related parameter one should assure that are such that the process gets activated. The ranges of the parameters and the parameters themselves were chosen based on a previous local sensitivity analysis.**

**FS:** - It is required to specify the scalar output used for sensitivity analysis, otherwise the results are meaningless. Sensitivity analysis results can strongly vary when considering different model outputs (e.g. van Werkhoven et al., 2008).

**SH: We agree that parameter sensitivities depend on the model output/ simulated process. Naturally one expects evaporation simulations to be sensitive to different parameters as streamflow simulations, particularly as these processes depend also in nature on different catchment characteristics. We will present a more accurate characterization of the 91 streamflow time series (as scalar model output) and input for FAST.**

**FS:** - Why analysing main effects (FAST) only and not total effects (eFAST)? A parameter could have an effect through interactions only and therefore would not be detected when applying FAST.

**SH: We agree with your statement and are aware of the shortcomings of FAST that solely covers the main effects of parameter changes on model output. Nevertheless, we considered the computationally efficient FAST method with a small number of factors (parameters) and interaction terms, thus smaller 'likelihood of nonnegligible higher-order terms' (Saltelli et al., 1999), as sufficient to illustrate the goals of the study. This reasoning is additionally supported following the main findings of Reusser et al. (2011), where the comparison of 3 different methods for Sensitivity analysis (including FAST and eFAST) yielded similar results in terms of sensitivities to model output. However we will better explain our choice in the revised manuscript.**

2) In Section 3.4.1 several points need clarification:

- p8 L3: 'sensitivity confined' is unclear.

**SH: Yes, sensitivity confined should somehow express that we relate or connect parameter sensitivity to the dynamic response of catchments either expressed by fingerprints, FDCs or hydrographs. We will clarify that statement by using a different expression.**

- p8 L4: this sentence is vague

**SH: We will clarify that one, too, or remove it from the manuscript.**

- p8 L5: this sentence have to be revised in relation to section 1.3.

**SH: We will better explain this part in connection to section 1.3.**

3) p9 L10-25: This is quite a long paragraph and I am not sure in which way it connects to the work presented in the manuscript. Again, I suggest to select the relevant information that helps to better understand the present study.

**SH: In the revision we will focus more precisely on regional parameter sensitivity linked to signatures (fingerprints) such as FDCs and their value to constrain the parameter space defined by FAST and their consistency in model performance and as control possibility of model adequacy. We will better illustrate the connections.**

SECTION 4 Study area

1) The authors should present the data used for the simulations and precise the simulation time horizon chosen for the analyses. Also, why presenting some of the data used before presenting the study area? (section 3.3.2, p7 L15-21)

**SH: The presentation of the data is indeed a point which has to be reworked and better structured.**

SECTION 5 Results

1) A general comments for this section is that some figures are redundant and could be removed for the sake of brevity (e.g. Figure 6, 9 and 10). Likewise, the text could be more concise.

**SH: Yes, as also stated in the answer to the 1[st] referee, we will reduce the number of figures to the minimum necessary amount and better specify captions.**

2) p11 L13: Why is it a 'qualitative constraint'?

**SH: Qualitative might be better removed here or at least replaced.**

3) p11 L29-30: The behavioural subset is the lower branch of the 'arrow' or the points that are within the circles?

**SH: We changed 'behavioural' to '...model runs with acceptable performance.' to avoid misunderstanding, since we only assign model runs that are within the circles to the group of behavioural parameter sets.**

4) p12 L2-3: What are the implications of the statements 'which portends [.. .] specifically.' and p12 L10-11 'Circles [. . .] varying strength'?

**SH: We will be more precise here and discuss the implications. The finding here is that local differences emerge between different catchments (though closely located) if the same constraint is used for the same parameter space in the vicinity of locations where behavioural parameter sets can be well selected while nearby places hardly produce any behavioural model run for the selected fingerprints.**

5) How does section 5.3 relates to section 5.1 which is also about constraint on parameter space?

**SH: Sorry for being unprecise here. Section 5.3 is the assessment of the performance of the behavioural parameter sets selected by constraints from observed fingerprints. We will better connect these two sections which build upon each other.**

6) In section 5.3, I think it may be more appropriate to identify for each fingerprint and pair of fingerprints not a unique most behavioural parameter set but an ensemble of best performing parameter sets. In fact, observation data are affected by uncertainties and it may not be relevant to make distinctions among the top performing parameter sets.

**SH: This is also a good point that we will further discuss and try to consider for the revision.**

7) Why keeping correlated fingerprints in the analysis of section 5.3? For instance, CV or HPC could be removed from the analysis since the two fingerprints have the same information content.

**SH: Yes, nevertheless this example might show how important it is to choose appropriate, non-redundant signatures. We will find a solution here.**

8) p14 L12: Please clarify how the five most behavioural parameter sets were determined.

**SH: We selected these 5 most behavioural parameter sets, by counting the number of times they occurred in the top five in reproducing the four combinations of fingerprints shown in fig. 5 and fig. 7c and d (referred to the seven shown headwaters). We will further clarify this selection.**

9) p15 L24-26: This statement is not correct. When referring to Sobol' variance decomposition (Sobol', 1990), the sum of main effects is equal to 1 when main effects only contribute to the total variance (no interactions). A sum of main effects for the six parameters analysed equal to 0.43 means that a significant fraction of the total variance (1-0.43=0.57) is due to interactions between parameters. If more parameters were added to the analysis, this would not necessarily result in an increase in the sum of the main effects, since the output variance would also change. The same applies for the statement p21 L11-12.

**SH: Thank you for this statement. We will check these parts and correct it.**

10) Section 5.4.1: I have reservations regarding the interpretation of the sensitivity analysis results: - p16 L4-10: I do not see any clear summer/winter pattern for the sensitivity of Aspect-corrPET and Recharge Coeff for WEN but more 'ups and downs' all year round.

**SH: Yes, the summer/winter pattern of AspectcorrPET and RechargeCoeff could be more pronounced but we can see a shift of the first rank sensitivity in the transition winter-summer from RechargeCoeff to AspectcorrPET and its generally higher average level of sensitivity in summer whereas RechargeCoeff doesn't change the average level of sensitivity significantly. We will try to clarify the interpretation.**

- p16 L26-35 - p17 L1-11: I am not sure about the significance of Figure 11. The authors consider the most influential parameter only. However, the difference in sensitivity among the most sensitive parameters may be small and even not statistically significant given the approximation error in the sensitivity index values. This is all the more concerning, since all the sensitivity indices take quite small values from Figure 9 (below 0.12).

**SH: We will find a way to present the relation of sensitivity to the independent variables (e.g. time or exceedance probability) in a more comprehensive way.**

11) p17 L10-11: The sentence 'Nevertheless [. . .] on stream flow generation.' is very fuzzy. It is required to clarify. I am also still wondering why the authors chose to study the main effects only and not the total effects.

**SH: We will clarify this sentence. Please also refer to our answer on eFAST above.**

SECTION 6 Discussions and Conclusions

1) I suggest splitting this section in two, with a discussion section and a concise conclusion section. This would help to highlight the contributions and implications of the work.

**SH: This is a good idea. Referee Björn Guse also recommended to split this section into two sub-chapters. We will incorporate this in the revised version of our manuscript.**

2) p18 L33: Either an indicator is normalised or it is not normalised but it cannot be 'less normalised'.

**SH: Correct. We changed it to '…more tangible and less biased (Krause et al., 2005) as it is e.g. in the case of NSE.'**

3) p19 L10-14 What are the implications of the statements 'The composition [. . .] in the eastern headwaters.'?

**SH: We agree to further clarify this point in terms of the implication for the performance analysis and parameter identification.**

4) p19 L18-19 'by model [. . .] of meteorological forcing data' is quite fuzzy. Please clarify.

**SH: Yes, meteorological forcing especially precipitation might have an effect on local differences in temporal parameter sensitivities. We will discuss this in more detail.**

5) p20 L22: The authors do not actually present any result on catchment classification

**SH: We apologize for setting the scope of the paper too wide. We will set a better focus on our main objectives.**

MINOR COMMENTS

- title of section 1.3: replace 'inconconsistency' by 'inconsistency'

**SH: We fixed this mistake.**

- there is an error in the reference to Beven (1990). Are the authors referring to Beven, K: Changing ideas in hydrology - The case of physically based models, Journal of Hydrology, 105, 157-172, doi: 10.1016/0022-1694(89)90101-7, 1989.

Or

Loague, K.: Changing ideas in hydrology - The case of physically based models - Journal of Hydrology, 120, 405–407, doi:10.1016/0022-1694(90)90161-P, http://linkinghub.elsevier.com/retrieve/pii/002216949090161P, 1990.

**SH: Fixed. The first reference is the one we intended to refer to.**

-p5 L8: replace 'of the selected fingerprints' by 'to the selected fingerprints'

**SH: Yes, fixed.**

-p5 L12 and L23: what do the authors mean by 'dependent'?

**SH: Dependent in the sense of the dependent variable streamflow in the case of FDCs and hydrographs. We will better explain these expressions.**

- p5 L13: replace 'of simulated stream flow and of the related fingerprints' by 'to simulated stream flow and to the related fingerprints'.

**SH: Yes, fixed.**

- p6 L21-23: Why introducing eFAST here if it is not used? I think the sentence can be removed to keep the manuscript concise, unless eFAST is actually used.

**SH: We agree with your recommendation**

-p8 L19: aren't there five dynamic fingerprints?

**SH: We incorporated the baseflow index as intermediate fingerprint in the dynamic ones to characterize low flow conditions.**

-p10 L14-15: it is required to reformulate this sentence.

**SH: We will reformulate this to make this statement clearer.**

- p14 L31: replace 'to small' by 'too small'

**SH: Yes, fixed.**

- p16 L25: I don't think the term 'interacting inversely' is correct since only the main effects (and not parameter interactions) are analysed here. Please reformulate.

**SH: We will reformulate this, given that main effects are analysed here.**

REFERENCES

Sarrazin, F., F. Pianosi, and T. Wagener. 2016. "Global Sensitivity Analysis of Environmental Models: Convergence and Validation." Environmental Modelling & Software 79: 135–52. doi:10.1016/j.envsoft.2016.02.005.

Sobol', I.M. 1990. "Sensitivity Estimates for Nonlinear Mathematical Models." Matematicheskoe Modelirovanie 2, 112-118 (in Russian), Translated in English (1993). In: Mathematical Modelling and Computational Experiments 1: 407–14.

Van Werkhoven, K., T. Wagener, P. Reed, and Y. Tang. 2008. "Characterization of Watershed Model Behavior across a Hydroclimatic Gradient." Water Resources Research 44 (1): 1–16. doi:10.1029/2007WR006271.

**SH: Thank you for this reference list. We will try to incorporate citations to these references in our revised manuscript.**

---

## Author Comment (AC4) · 16 Aug 2016

Response to Anonymous Referee #4

**Anonymous Referee (AR)**: I have to admit that I struggled for a couple of days to understand the message of this paper, and I am disappointed to say I failed to do so. My understanding is this paper is an amalgam of sensitivity analysis, parameter estimation and catchment clustering, however it is not well described how these approaches are linked together. Each of these elements, if performed elaborately, can be a separate paper and mixing them only confuses readers. Neither the abstract nor the introduction sections support the goals of the study. Well to be fair, goals are not clear either! Also I should mention that manuscript is not fluent at times. In my comments, I only focus on major flaws of the manuscript and skip my minor comments:

**Simon Höllering (SH)**: **We thank the referee for his critical assessment of our manuscript.**

**AR**: I noticed there are several unsupported claims in the manuscript, one of them is "parameter estimation". I can't find how the parameter estimation is performed. It is not close to sufficient to consider the 91 parameter combinations from the sensitivity analysis and use a selection criteria based on some fingerprints of the catchment to select parameters as behavioral from this limited set. This would lose many behavioral parameter combinations, and doesn't provide any information about the posterior distribution.

**SH: Yes, as stated in the reply to referee 1, we are aware that a number of 91 runs is certainly too small to find parameter sets which work well for most of the fingerprints. Nevertheless, we found several to work fine for specific fingerprints and some that are not far from the results (in terms of the NSE) we obtained by automatic DDS calibration (0.75 vs. 0.81). In accordance with a statement by referee Björn Guse we will mention in the revised manuscript that this lower number of model runs is acceptable here to show the goals of this study, which is different from automated model calibration.**

**AR**: I might be wrong, but my understanding is that the sensitivity analysis of model parameters depends on a residual based objective function. At least it is highly dependent on simulation of the system response at individual time steps! This is in contrast with the purpose of using fingerprints of catchments, which are originally defined to constrain model parameters to represent an aggregate behavior of the catchment.

**SH: Sensitivity analysis using FAST rests upon the idea that model simulation output can be transferred into a Fourier series. Within feasible, predefined ranges, parameters are varied with specific, independent frequencies. The Fourier coefficients finally allow an estimation of the partial variance, hence model parameter sensitivity, used to identify dominant parameters. In this process of sensitivity analysis an ensemble of model runs (based on different parameter combinations) is required (91 in our case of 6 parameters) which we also used for the assessment of model performance and consistency check in terms of the reproduction of response fingerprints. Given the implementation of FAST we do not need an objective function here (such as NSE or MSE). Similarly, as responded to referee 3 and what we will clarify in the revised manuscript, a structurally adequate model with behavioral parameters sets should consistently allow for acceptable stream flow simulations and reproduce fingerprints derived from stream flow. An inconsistent behavior in the sense that parameters which work well during stream flow simulations, but do not work well with respect to reproduce fingerprints provides evidence for model structural error.**

**AR**: It is not clear how the selection criteria is adopted to delineate the behavioral parameter distribution. There are times that authors discuss one fingerprint is used, whereas in other instances they used a couple of fingerprints jointly!

**SH: Sorry for being imprecise, we will clarify the selection of behavioural parameter sets in the revised version of the manuscript and better discuss the use and value of both one and several fingerprints as constraining measures in separate sections.**

**AR**: In the original application of fingerprints that authors referred to (Vrugt and Sadegh, 2013), 4 fingerprints were uses that are necessary to meet the acceptance criteria jointly. It is not clear if authors have performed their analysis on single sites (headwaters), or they have modeled the entire system altogether.

**SH: The simulation was run and sensitivity analysis was performed on each single headwater and, as we showed in several figures, fingerprints were separately derived. Five most behavioural sets were in the end chosen jointly from seven headwaters (where the overall model performance was acceptable) and further assessed.**

**AR**: Page 2, line 32: My experience shows that, at least for US catchments, parameters of certain models are more correlated with climatic variables rather than soil characteristics. It is worth mentioning here, although the sentence is correct in how it describes the findings.

**SH: We agree with this statement. This section might be removed, thus catchment classification will not be relevant for the revised manuscript.**

**AR**: In section 3.3.2, authors talk about the study area before introducing it!

**SH: Yes, we will correct this point.**

**AR**: Page **8**, Line 3: I don't understand the sentence: "relate physiographic and climatic characteristics to sensitivity-confined hydrodynamic response fingerprints"

**SH: We apologize for the unprecise formulation. In fact we try to relate the connection of regional differences in hydrologically relevant landscape properties and climatic factors (which have shown here to be heterogeneous even in nearby locations) and the hydrologic response expressed by fingerprints which can be further related to model parameters which might be sensitive to a fingerprint. This insight can be e.g. used to diagnose model structural deficits.**

**AR**: Page 10, line 7: Water is withdrawn from the system, it is not lost!

**SH: Yes, fixed this text passage.**

**AR**: Section 5.3: authors talk about consistency of behavioral parameter sets. In what sense have you analyzed the consistency of parameter sets? For definition of hydrologic consistency refer to: Martinez, G. F., and H. V. Gupta (2011), Hydrologic consistency as a basis for assessing complexity of monthly water balance models for the continental united states, Water Resources Research, 47 (12).

**SH: Martinez, G. F., and H. V. Gupta (2011) point out that 'the term consistency can mean a great many different things, including what might be called (1) internal consistency with regard to the conceptual representation of a model (…) and (2) external consistency in terms of the ability to reproduce hydrological behaviors and properties seen in the data.' In our case, the evaluation of behavioural parameter sets was done with respect to their consistency in reproducing one single or two combined fingerprints of hydrodynamic response. This might be most similar to what is meant by external sensitivity.**

**AR**: Page 15, line 25: In the entire manuscript authors are talking about 6 model parameters, and all of a sudden they switch to 52 global mHM parameters! It confuses me which one is the correct number of model parameters.

**SH: We changed this part to better highlight that we used 6 parameters out of the whole set of 52.**

**AR**: Page 16, line 26: Authors suddenly talk about temporal sensitivity of parameters! This is completely different from what reader expect from a joint sensitivity-parameter estimation analysis. The latter works with the entire data set, whereas the former is concerned about individual time steps!

**SH: In accordance to the response to referee comment 3 we will clarify objectives e.g. in the way that we extend the TEDPAS (temporal dependence of parameter sensitivity) concept to fingerprints (FDCs, HFD, etc.) to depict changes in their parameter sensitivity for changes of the independent variable INDPAS and optionally assess regional differences.**

**AR**: Page 19 lines 6-16: Categorizing catchments based on model parameter assumes that the model is sufficiently describing the system. This assumption is not well justified nor supported by the results.

**SH: We agree with the reviewer that any model based categorization of catchments is based on the assumption that the model is structurally adequate. Here we do not categorize the catchments on parameters, but we compared differences TEDPAS and the partial sensitivities among catchments. Differences in the sensitivities underpin the differences in parameter identifiability in these catchments. As parameter ranges of the FAST are the same within all these catchments. These differences may be attributed to differences in the hydro-climatic input. We will better explain this in the revised manuscript.**

**AR**: Page 20, lines 21-22: This is again unjustified claim to say this paper does: "(1) investigate hydrologically relevant structural and functional attributes in terms of consistency and feasibility in classifying similar catchments, (2) assess the value of functional constraints for the parameter spaces of distributed hydrologic models". I am not convinced that the results of this study support these claims.

**SH: We will remove these passages from the revised manuscript, as catchment classification is not within the scope of this study.**

**AR**: Figure 3 is not well explained in the text.

**SH: Yes, we think about removing this figure in the revised version.**

**AR**: Figure 10 & 11: How is it possible to differentiate between model simulations of the two gauges?

**SH: The simulations here are only shown for gauge Wenholthausen (WEN) but observations for both (with the reference to parameters of highest sensitivity). To be clearer here we will add the names of gauges to the curves.**

**AR**: Figure 12: Why four of the parameters and not all 6?

**SH: Good, we also think about showing SDCs of all the six parameters.**